# Molecular mechanisms of thiazide-like diuretics-mediated inhibition of the human Na-Cl cotransporter

Chien-Ling Lee [1,2], Jianxiu Zhang [1,2] & Liang Feng [1] ✉

Thiazide-type and thiazide-like diuretics are structurally distinct first-line antihypertensive drugs that target the sodium-chloride cotransporter (NCC) in the kidney. Thiazide-like diuretics are reported to have better cardioprotective effects than thiazide-type diuretics, but whether this is due to differences in NCC-inhibition mechanisms, if there is any, remains unclear. To understand the molecular mechanisms of NCC inhibition by thiazide-like diuretics, we determine the structures of human NCC (hNCC) bound to two of the most widely used thiazide-like diuretics, chlorthalidone and indapamide, using cryogenic electron microscopy (cryo-EM). Structural analyses reveal shared features and distinctions between NCC-inhibition by thiazide-like and thiazide-type diuretics. Furthermore, structural comparisons allow us to identify polymorphisms in hNCC that have substantial differential effects on the potencies of specific thiazide-like and thiazide-type diuretics. Our work provides important insights into the molecular pharmacology of NCC and a blueprint for developing precision medicine to manage hypertension with thiazide-like and thiazide-type diuretics.

Hypertension (elevated arterial blood pressure) is a widespread and serious medical condition that affects approximately one-third of the global adult population[1,2]. It causes damages to multiple organs[3,4] and represents a leading risk factor for premature death worldwide[1,5]. It is well established that pharmacological agents that lower blood pressure are essential to reducing hypertension-related morbidity and mortality[6–9]. Thiazide-type and thiazide-like diuretics (collectively, thiazide diuretics in this paper) are first-line antihypertensive drugs[10–13] used to treat 20–40% of hypertensive patients[14–18]; they have served as a cornerstone for managing hypertension for decades[19].

Pharmacodynamically, thiazide-type and thiazide-like diuretics cause similar diuretic effects[20,21] by inhibiting the NCC transporter in the distal convoluted tubule (DCT) of the kidney[22]. Thiazide-like diuretics are structurally distinct from thiazide-type diuretics, however, because they lack the benzothiadiazine scaffold shared by thiazide-type diuretics[23,24] (Supplementary Fig. 1). For example, chlorthalidone (CTN) and indapamide (IDP) are phthalimidine and indoline

derivatives of benzenesulfonamide, respectively[21,23] (see below). Although still under debate[19,25], several studies found that thiazide-like diuretics have better cardioprotective effects than thiazide-type diuretics[26], leading to preferential recommendations for thiazide-like diuretics over thiazide-type diuretics in several hypertension guidelines[10,12,27,28].

A patient's genetic profile also can have a major impact on their sensitivity and responsiveness to thiazide diuretics: genetic variation is a major contributor to inter-individual variability in response to specific antihypertensive drugs[29–31]. For thiazide diuretics, the profound effect that genetic variation can have on individuals' responses to specific drugs has been documented[32–35], yet how genetic variations exert their influence has not been explained, nor has inter-individual variability in response to different thiazide diuretics been characterized. Although thiazide diuretics are considered interchangeable clinically[19], considerable structural differences among thiazide-type and thiazide-like diuretics suggest they may engage in distinct, specific

[1]Department of Molecular and Cellular Physiology, Stanford University School of Medicine, Stanford, CA, USA. [2]These authors contributed equally: Chien-Ling Lee, Jianxiu Zhang. ✉e-mail: liangf@stanford.edu

interactions with NCC, raising the possibility that particular thiazide diuretics may work best for managing patients with specific genetic variations of NCC. A major barrier for identifying such variants with important pharmacological implications is the lack of molecular understanding of common classes of thiazide diuretics.

Recently, the structure of NCC bound to a thiazide-type diuretic, polythiazide, provided important insights into how this group of thiazide diuretics inhibit NCC[36]. Although informative, our knowledge gap about the underlying molecular mechanisms of NCC inhibition by thiazide-like diuretics persists. In particular, it remains unclear whether thiazide-like diuretics inhibit NCC through a mechanism similar to thiazide-type diuretics. Furthermore, thiazide-like diuretics have diverse structures. How these structurally distinct molecules bind to NCC and inhibit its transport function remains a key mechanistic question. The answers to these questions are crucial for understanding the actions of thiazide-like diuretics and dissecting their potential pharmacological benefits compared with thiazide-type diuretics.

In this work, we carry out structural and functional studies to investigate the mechanisms of thiazide-like diuretics-mediated inhibition of NCC. We focus on two clinically significant thiazide-like diuretics, chlorthalidone and indapamide, which are the most prescribed thiazide-like diuretics in the US[37] and UK[38], respectively.

## Results

### Structural determination

Given that chlorthalidone, indapamide, and thiazide-type diuretics can inhibit metolazone binding to NCC[39], we reasoned that their binding sites may overlap and likely form in an outward-facing conformation, as revealed in the polythiazide-bound NCC structure[36]. The predominantly inward-facing conformation adopted by NCC in detergent micelles[36] would thus be incompatible with binding these molecules. To address this, we first adopted a strategy that previously enabled structural determination of polythiazide-bound hNCC[36]: substituting the N-terminal domain (NTD) with that of the zebrafish NKCC1 (DrNKCC1) and incorporating an extracellular gate mutation, E240A, that favors an outward-facing conformation (the construct is termed NCC$_{cryo}$). NCC$_{cryo}$ is functional (albeit with lowered activity) and has similar affinities to chlorthalidone- and indapamide-mediated inhibition as wild-type hNCC (Fig. 1f, g and Supplementary Fig. 2c). However, cryo-EM studies revealed that detergent samples of NCC$_{cryo}$ copurified with chlorthalidone or indapamide yielded only blurry two-dimensional (2D) class averages. We suspected this might be due to the substantially lower apparent affinities of indapamide and chlorthalidone compared to polythiazide (~16 and ~40 times lower, respectively) (Supplementary Fig. 2d), which might reduce their ability to stabilize the outward-facing conformation. Recognizing that thiazide-like diuretics can bind to and inhibit NCC in the plasma membrane, we reasoned that a more native-like membrane environment, such as nanodiscs, might help stabilize the chlorthalidone- and indapamide-bound structures. Using NCC$_{cryo}$ reconstituted in nanodiscs, we determined a cryo-EM structure of indapamide-bound hNCC at 2.79 Å resolution and two cryo-EM structures of chlorthalidone-bound hNCC at 3.01 Å and 3.24 Å resolution, respectively (Fig. 1a–c, Supplementary Figs. 2a,b, 3, and 4, and Supplementary Table 1).

### Overall architecture of indapamide- and chlorthalidone-bound hNCC

In all structures of hNCC in complex with indapamide and chlorthalidone, hNCC exists as a homodimer with the transmembrane domain (TMD) and C-terminal domain (CTD) in a domain-swapped configuration (Fig. 1a–c), sharing similar overall architecture with the polythiazide-bound structure[36].

Intriguingly, we observed differences in domain-domain interactions among these structures. In the indapamide-bound hNCC structure, two protomers form a symmetrical dimer, in which both CTD subunits are bound with an NTD and form close interactions with the TMD (Fig. 1a, c). This contrasts with the polythiazide-bound structures (Fig. 1c and Supplementary Fig. 5e), in which only one copy of the CTD is bound with an NTD and forms similar close interactions with the TMD. This results in notable difference in overall dimer organization between the indapamide- and polythiazide-bound hNCC (Protein Data Bank (PDB) ID: 8FHN) structures, with a root mean square deviation (RMSD) of 7.29 Å for the NCC dimer, despite highly similar individual domain structures. Overall, the NTD-CTD interaction mode is comparable in these structures (Supplementary Fig. 5c). Compared with the polythiazide-bound structure, the densities of the NTD (from DrNKCC1) in the indapamide-bound structure are discernibly weaker, particularly for residues that are not directly involved at the NTD-CTD interface (Supplementary Fig. 5a). This suggests the NTD is potentially more flexible, and the NTD-CTD interactions might be less stable. In addition, in the indapamide-bound hNCC structure, the two TMDs are well separated and do not form the direct interactions observed in the polythiazide-bound structure (Fig. 1c–e). The distance between the two TMD subunits increases by ~17 Å compared to the polythiazide-bound structure (Fig. 1e). In association with these different TMD-CTD and TMD-TMD interaction modes, the relative rotation between the TMD and CTD dimers is greater in the indapamide-bound structure than in the polythiazide-bound one (Fig. 1d).

For chlorthalidone-bound hNCC, two structures with different dimer configurations were determined (Fig. 1b, c). One structure (3.24 Å resolution) has overall asymmetrical architecture similar to that of polythiazide-bound (PDB: 8FHN) hNCC (RMSD = 0.978 Å), with only one NTD bound to a CTD subunit, and only this CTD subunit makes close interactions with a TMD subunit (Fig. 1b, c and Supplementary Fig. 5g). Interestingly, in this asymmetrical structure, the other TMD subunit is less well resolved, especially the extracellular cap domain, compared with the well resolved TMD dimers in the polythiazide-bound structures (Fig. 1b and Supplementary Fig. 5d). Another chlorthalidone-bound structure (3.01 Å resolution) adopts a symmetrical architecture similar to that of indapamide-bound hNCC (RMSD = 0.474 Å) (Fig. 1b, c and Supplementary Fig. 5f). The presence of two different dimer configurations and a less well resolved TMD subunit in the asymmetrical structure suggest potentially greater intradimer dynamics in chlorthalidone-bound hNCC. This is consistent with 3D variability analysis, which helped to better resolve the less CTD-interacting TMD subunit of the asymmetrical structure (Supplementary Fig. 5d). Similar to the indapamide-bound state, the discernibly weaker NTD densities in both symmetrical and asymmetrical structures (Supplementary Fig. 5a), despite comparable NTD-CTD interaction modes (Supplementary Fig. 5b), suggest that the CTD-interacting region of the NTD might also be more flexible in the chlorthalidone-bound state than in the polythiazide-bound state.

### Indapamide- and chlorthalidone-binding sites

In all indapamide- and chlorthalidone-bound hNCC structures, both TMD subunits adopt similar outward-facing conformations with a vestibule lined by TM helices 1, 3, 6, 8, and 10. This vestibule opens wide to the extracellular environment and extends to the middle of the membrane, similar to the polythiazide-bound hNCC (Figs. 2b, 3a, and 6a). In both TMD subunits, we found bulky non-protein densities that fit indapamide and chlorthalidone, respectively, in the vestibule's intracellular ends (Figs. 2b–d and 3a–c).

Both indapamide and chlorthalidone bind hNCC with their sulfamoyl and chlorine groups near the bottom of vestibule, and indapamide's indoline moiety and chlorthalidone's phthalimidine moiety point extracellularly (Figs. 2b, e and 3a, d). Indapamide, chlorthalidone, and polythiazide share a common moiety (Supplementary Fig. 1), benzenesulfonamide, that shows similar binding poses and forms comparable interactions with hNCC based on the structures (Figs. 4a and 6c). This moiety stacks with F536 via π–π interactions

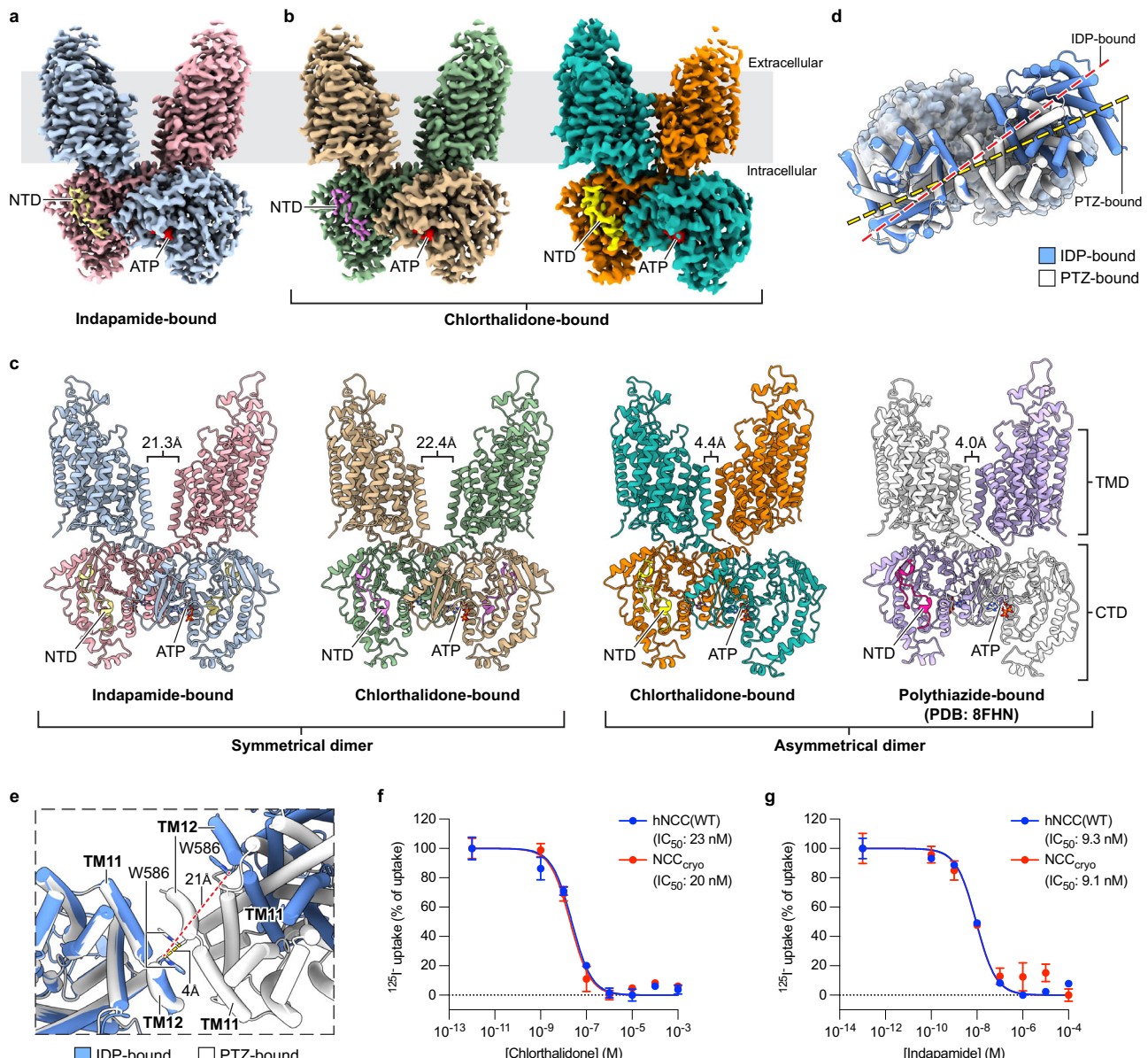

**Fig. 1 | Overall architecture of indapamide-bound and chlorthalidone-bound hNCC. a, b** Cryo-EM density maps of indapamide (IDP)- and chlorthalidone (CTN)-bound NCC$_{cryo}$. The IDP-bound and symmetrical and asymmetrical CTN-bound maps are contoured at 4.3 σ, 5.1 σ, and 4.3 σ, respectively. The two protomers and NTD of the IDP-bound structure are colored light blue, pink, and pale yellow, respectively. The two protomers and NTD of the symmetrical CTN-bound structure are colored tan, green, and violet, respectively. The non-NTD-bound and NTD-bound protomers and NTD of the asymmetrical CTN-bound structure are colored teal, orange, and yellow, respectively. All ATP/ADP densities are colored red. **c** Ribbon representations of the structures of IDP-, CTN-, and polythiazide (PTZ)-bound hNCC. The color schemes of IDP- and CTN-bound structures are the same as in (**a**). The NTD-bound and non-NTD-bound protomers and NTD of the PTZ-bound structure are colored light purple, white, and deep pink, respectively. Bound ATP/ADP molecules are shown as color-matched sticks. The distances between TMD subunits (the distance between the Cβ of W586 pair) are labeled. **d, e** Ligand-specific hNCC dimer configurations. The IDP- and PTZ-bound structures are colored blue and white, respectively. The TMD and CTD are shown as cylinder and surface representations, respectively, and the cap domains are not shown for clarity. The structural alignment is based on TMD subunits forming extensive TMD-CTD interactions. In **e** the distances between the two TMD subunits are outlined with red and yellow dashed lines for IDP-bound and PTZ-bound hNCC structures, respectively. **f, g** NCC$_{cryo}$ sensitivity to chlorthalidone and indapamide. Data are shown as mean ± SEM ($n = 3$ independent experiments). The baseline of the thiazide-sensitive iodide uptake is marked by a dotted line.

(Figs. 2e and 3d). Notably, its sulfamoyl group interacts with N149 and N227 via hydrogen bonds (Figs. 2e and 3d). Based on the cell-based functional assay, alanine substitution for N227 resulted in over 10,000- and 1000-fold increases in the half-maximal inhibitory concentrations (IC$_{50}$) of indapamide and chlorthalidone, respectively (Figs. 2f, 3e, f, and Supplementary Fig. 9c). Likewise, the N149A substitution made hNCC approximately 27- and 95-fold less sensitive to indapamide- and chlorthalidone-mediated inhibition, respectively (Figs. 2f and 3e, f, and Supplementary Fig. 9c). These results are consistent with N149 and N227's important roles in binding these thiazide-like diuretics. Interestingly, the relative impact of alanine substitutions on potency varies substantially among the drugs; for example, N227A substitution affects indapamide over 50-fold more than the previously reported polythiazide[36], suggesting other parts of the drug molecules make differential contributions to binding hNCC.

For both indapamide and chlorthalidone, a chlorine group ortho to the sulfamoyl group (Figs. 2d and 3c) occupies the Cl⁻-binding site that was previously identified in inhibitor-free NCC structures[36,40] and

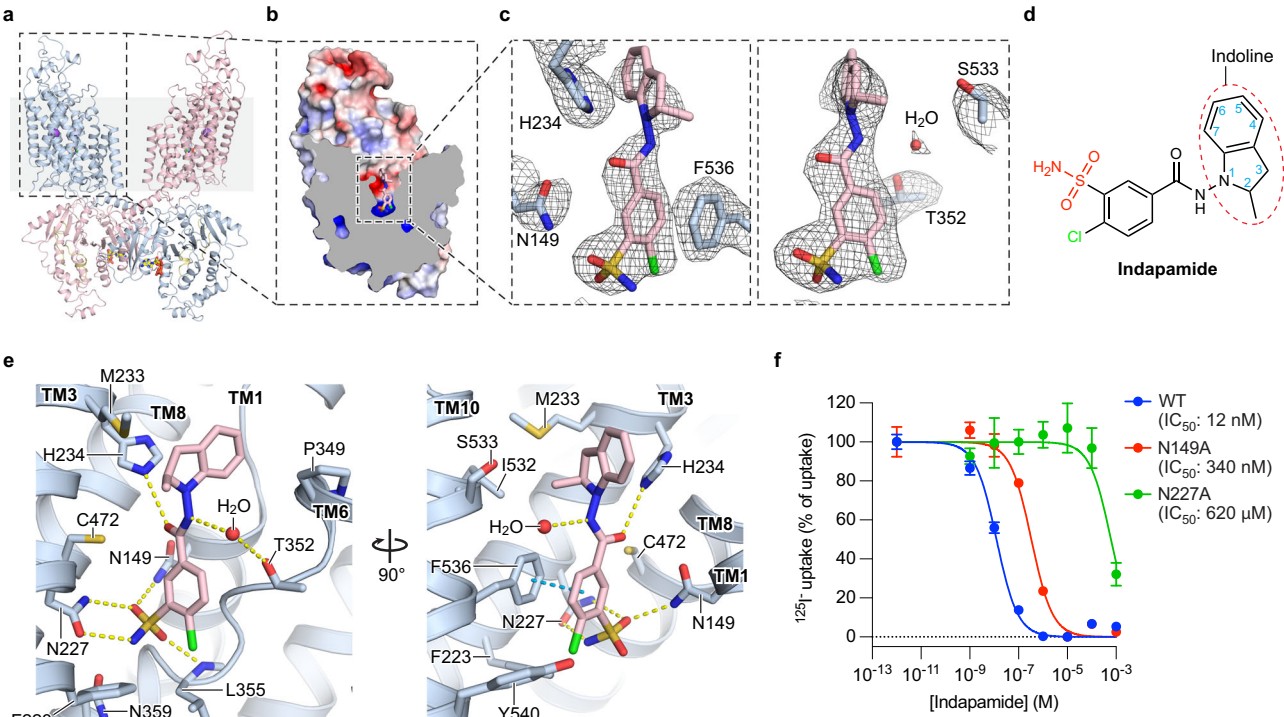

**Fig. 2 | Indapamide-binding site. a** Ribbon representation of indapamide (IDP)-bound NCC_cryo. The two protomers and NTD are colored light blue, pink, and pale yellow, respectively. The bound indapamide and ATP molecules are displayed as sticks, colored light blue/pink and yellow, respectively. The bound Na⁺ ions are shown as purple spheres. **b** Sliced view of one TMD subunit of IDP-bound NCC_cryo. The protein surface representation is colored by electrostatic potential, in range of red (−5 kT e⁻¹) to blue (+5 kT e⁻¹), and the bound indapamide molecule is shown as sticks. **c** Densities of indapamide and nearby residues. Indapamide and nearby selected residues are shown as pink and light blue sticks, respectively, and the densities are shown as meshes. The contour levels of the left and right panels are 9.5 σ and 6.5 σ, respectively. **d** Chemical structure of indapamide. The sulfamoyl and chlorine groups are colored red and green, respectively. The indoline moiety is outlined with a red dashed oval, and the numbering system of the indoline moiety is shown as blue numbers. **e** Indapamide-hNCC interactions. Hydrogen bonds are shown as dashed lines in yellow, and π–π stacking interactions in cyan. The color scheme is the same as in (**c**). **f** Effects of alanine substitutions of key IDP-interacting residues on hNCC sensitivity to indapamide. Data are shown as mean ± SEM (n = 3 independent experiments).

interacts with Cl⁻-coordinating residues (Fig. 6c). Thus, they directly prevent Cl⁻ binding. Consistent with this, we did not observe ion-like density in the Cl⁻-binding site. Removing the chlorine group from chlorthalidone nearly abolished diuretic activity[41], corroborating this group's critical role in drug function. In contrast to the Cl⁻-binding site, the Na⁺-binding site of hNCC is outside the inhibitors' binding region and is occupied by a sodium ion in the indapamide-bound or symmetrical chlorthalidone-bound structure (Supplementary Fig. 7a,b). In the asymmetrical chlorthalidone-bound structure, the Na⁺-coordinating residues adopt similar structures as the Na⁺-bound indapamide-bound and symmetrical chlorthalidone-bound structures (Supplementary Fig. 7c), suggesting a Na⁺-bound state although ion density cannot be unambiguously identified in the Na⁺-binding site. The Na⁺-binding sites of the indapamide- and chlorthalidone-bound structures align well with that of the polythiazide-bound structure (Supplementary Fig. 7c), suggesting similar effects of extracellular sodium ions on both types of thiazide diuretics. Similar to Na⁺ binding, both nucleotide-binding sites in the CTD are bound with ATP/ADP molecules in all indapamide- and chlorthalidone-bound hNCC structures, the same as the polythiazide-bound structures (Supplementary Fig. 7d–g), suggesting similar nucleotide binding states in the CTD.

In contrast to the benzenesulfonamide moiety, which adopts a similar pose across all inhibitors studied, indapamide's indoline moiety and chlorthalidone's phthalimidine moiety adopt unique binding poses.

Indapamide's indoline moiety, consisting of fused benzene and nitrogen-containing five-membered rings (Fig. 2d), is oriented with its plane perpendicular to the membrane plane and the 2-position carbon of the five-membered nitrogen-containing ring pointing toward the

interface between TM3 and TM10 (Fig. 2e). Its indoline and benzene-sulfonamide moieties are connected through an amide bond, whose carbonyl and amine groups form hydrogen bonds with the side chain of H234 and a water molecule, respectively (Fig. 2c, e). This interacting water molecule also forms polar contacts with the side chains of T352 and S533 (Fig. 2c, e). Thus, unlike polythiazide, which interacts directly with T352, indapamide interacts indirectly with T352 through a water molecule. In addition, the indoline moiety forms π–π stacking interactions with H234 (Fig. 2c, e) and makes hydrophobic interactions with M233, P349, and I532 (Fig. 2e). Given the indoline moiety occupies an open space that leads to the extracellular solution, this explains the finding that the binding site can accommodate indapamide derivatives with differently oriented bicyclic groups. For example, an indapamide derivative with the indoline moiety replaced with an isoindoline, where its bicyclic group is oriented perpendicular to that of indapamide, has diuretic activity similar to indapamide[42].

Chlorthalidone's phthalimidine moiety also has a bicyclic structure consisting of fused benzene and nitrogen-containing five-membered rings (Fig. 3c). The direct connection between the phthalimidine moiety and the benzene ring of the benzenesulfonamide moiety makes the long axis of chlorthalidone shorter than that of indapamide (Fig. 4a). The phthalimidine moiety is oriented with the 2-position nitrogen of the five-membered ring pointing toward the TM1/6 interface, opposite to the orientation of indapamide's indoline moiety (Figs. 3c, d and 4a). In addition, the phthalimidine moiety forms a ~60° angle with the membrane plane, differing from the perpendicular orientation of indapamide's indoline moiety (Figs. 3a, b, d and 4a). The phthalimidine moiety makes multiple interactions with NCC

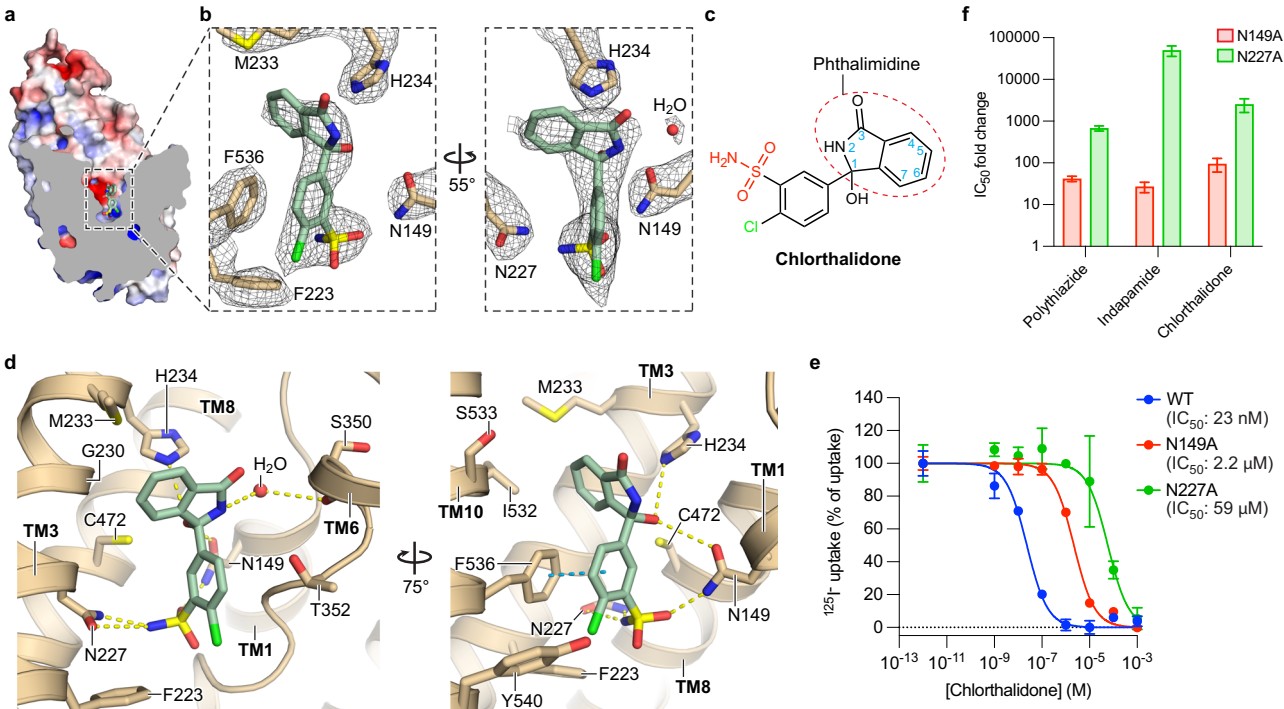

**Fig. 3 | Chlorthalidone-binding site. a** Sliced view of one TMD subunit of chlorthalidone (CTN)-bound NCC$_{cryo}$. The surface representation of the protein is colored by electrostatic potential, in range of red ($-5$ kT e$^{-1}$) to blue ($+5$ kT e$^{-1}$). The bound chlorthalidone molecule is shown as sticks. **b** Densities of chlorthalidone and nearby residues. hNCC and chlorthalidone are colored in tan and green, respectively. The contour levels of the left and right panels are 9.5 σ and 7.5 σ, respectively. **c** Chemical structure of chlorthalidone. The sulfamoyl and chlorine groups are colored red and green, respectively. The phthalimidine moiety is outlined with a red dashed oval, with the numbering system shown as blue numbers. **d** Chlorthalidone-NCC interactions. Hydrogen bonds are shown as dashed lines in yellow, and π–π stacking interactions in cyan. The color scheme is the same as in (**b**). **e** Effects of alanine substitutions of key CTN-interacting residues on hNCC sensitivity to chlorthalidone. Data are shown as mean ± SEM ($n = 3$ independent experiments). Data for WT are same as in Fig. 1f. **f** Differential effects of N149A and N227A substitutions on hNCC sensitivities to several thiazide diuretics. The bar heights are best-fit values of IC$_{50}$ fold change relative to hNCC(WT) calculated from Figs. 2f (indapamide) and 3e (chlorthalidone) using the EC50 shift analysis in GraphPad Prism 10. For polythiazide, best-fit values were calculated from sensitivities reported in ref. 36. The error bar depicts the standard error reported by GraphPad Prism 10, which serves as an indicator of the confidence of calculated IC$_{50}$ fold change.

(Fig. 3b, d), including hydrogen-π interactions with H234 in a T-shaped orientation to its imidazole side chain[43,44], hydrogen bonds with the side chains of H234 and N149 through its 1-position hydroxyl group, and hydrophobic interactions with G230, M233, and I532. Except direct interactions, the 2-position amine (NH) group of the phthalimidine moiety also forms indirect interactions with the main-chain oxygen of S350 through a water molecule (Fig. 3b, d). In the two chlorthalidone-bound structures with different dimer configurations, chlorthalidone adopts very similar binding poses and forms very similar protein-ligand interactions (Supplementary Fig. 6). Therefore, unless otherwise specified, the symmetrical chlorthalidone-bound structure will be used for discussion, given its higher resolution.

**Ligand-specific interactions**

Regions outside the shared benzenesulfonamide moieties of indapamide, chlorthalidone, and polythiazide form distinct interactions with several residues of hNCC, including G230, H234, T352, C472, and S533. G230 forms hydrophobic interactions with chlorthalidone but not indapamide or polythiazide. H234 forms a hydrogen bond with the 1-position sulfonyl (SO$_2$) group of polythiazide. In comparison, the H234 side chain forms much stronger interactions with the indoline and phthalimidine moieties of indapamide and chlorthalidone, respectively: not only does it engage in hydrogen bonding, the H234 side chain also forms π–π stacking interactions and hydrogen-π interactions with the indapamide's indoline and chlorthalidone's phthalimidine moieties, respectively (Figs. 2e and 3d). The T352 side chain forms a hydrogen bond directly with the 4-position amine (NH)

group of polythiazide but interacts indirectly with indapamide, mediated by a water molecule (Fig. 2e); in contrast, T352 does not form polar interactions with chlorthalidone. For C472, its side chain forms polar interactions with either oxygen of the 1-position sulfonyl (SO$_2$) group of polythiazide (Supplementary Fig. 8d). In contrast, the side chain of C472 forms polar interactions with the 1-position hydroxyl group of chlorthalidone's phthalimidine moiety and no polar interaction with indapamide (Supplementary Fig. 8e, f). The potentially stronger interaction between C472 and polythiazide is consistent with the strong densities connecting the side chain of C472 and the 1-position sulfonyl (SO$_2$) group of polythiazide in the cryo-EM map, a feature not seen in the chlorthalidone- and indapamide-bound structures (Supplementary Fig. 8a–c). The side chain of S533 forms hydrophobic interactions with polythiazide, but it interacts with indapamide indirectly through a water molecule. In contrast, S533 does not appear to be within range to interact with chlorthalidone. Interestingly, S533's side chain points toward the bound indapamide to form water-mediated interaction, but it rotates and points away from polythiazide to accommodate its 3-position -CH$_2$-S-CH$_2$-CF$_3$ moiety (Fig. 5a and Supplementary Fig. 9b). Compared to the indapamide-bound structure, TM10 of the less CTD-interacting TMD subunit of the polythiazide-bound structure shows a clockwise rotational shift when viewed from the dimer interface (Fig. 5b). Because the intracellular loop 5 (IL5) and TM11/12 following TM10 are involved in TMD-CTD interactions and the dimer interface, respectively, conformational differences in TM10 may contribute to the differential NCC dimer configurations in the polythiazide- and indapamide-bound structures

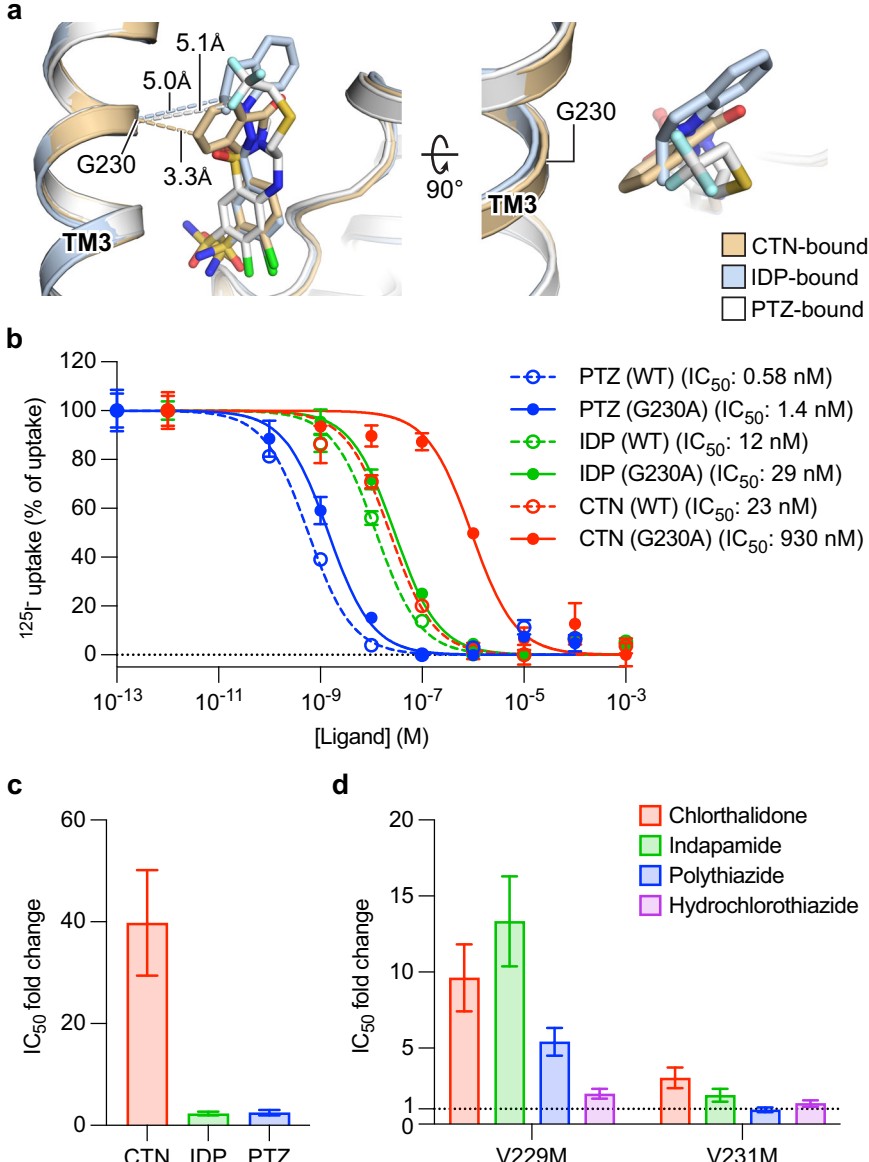

**Fig. 4 | Ligand-specific interactions with hNCC. a** Interactions between G230 and several thiazide diuretics. The CTN-, IDP-, and PTZ-bound structures are colored tan, light blue, and white, respectively. The three structures are aligned based on TM3. The closest distances between the Cα of G230 and the three molecules are outlines with color matched dashed lines. **b** Dose-response curves of several thiazide diuretics against hNCC(WT) or hNCC(G230A). Curves for hNCC(WT) and hNCC(G230A) are shown as dashed and solid lines, respectively. Curves for chlorthalidone, indapamide, and polythiazide are colored red, green, and blue, respectively. Data are shown as mean ± SEM (*n* = 3 independent experiments). Data for IDP(WT) and CTN(WT) are the same as in Figs. 2f and 1f, respectively. **c** Differential effects of G230A substitution on hNCC sensitivities to several thiazide diuretics. The bar heights stand for best-fit values of $IC_{50}$ fold change relative to hNCC(WT) calculated from Fig. 4b using the EC50 shift analysis in GraphPad Prism 10. **d** Differential effects of V229M and V231M polymorphisms on hNCC sensitivities to different thiazide diuretics. The bar heights are best-fit values of $IC_{50}$ fold change relative to hNCC(WT) calculated from Supplementary Fig. 9g (V229M) and 9 h (V231M) using the EC50 shift analysis in GraphPad Prism 10. The level of $IC_{50}$ fold change equal to 1 is outlined with a dashed line. In **c**, **d** the error bar represents the standard error, which serves as an indicator of the confidence of $IC_{50}$ fold change calculation by GraphPad Prism 10.

(Fig. 5c). Given its location in the extracellular end of TM10, the ligand-dependent interactions with S533 might underlie the differential dimer configurations between polythiazide- and indapamide-bound hNCC structures. For chlorthalidone, because of the lack of interaction with S533 and the longer distance to the extracellular end of TM10, it probably has a neutral effect on the local conformation of the extracellular end of TM10 (Fig. 5a and Supplementary Fig. 9a). This might be the reason why chlorthalidone-bound hNCC can adopt either dimer configuration similar to the polythiazide-bound or indapamide-bound structure (Fig. 5b, c).

The distinct interactions of the three thiazide diuretics with these residues suggest that their mutations may differentially impact

NCC's sensitivity to various thiazide diuretics. However, residues H234, T352, C472, and S533 are essential for NCC transport function: single variants of these residues (H234A, T352A, C472A, and S533T) severely impaired hNCC transport function (Supplementary Fig. 9d), making it impractical to assess inhibitors' potencies to these NCC variations. Thus, we focused on G230, which is in range to contact chlorthalidone but not indapamide or polythiazide, with the closest distances being 3.3 Å, 5.0 Å, and 5.1 Å, respectively (Fig. 4a). These distances suggest substituting G230 with bulkier residues might differentially affect hNCC's ability to bind these three molecules. Indeed, we found that substituting alanine for G230, which has a minimal effect on transport activity (Supplementary Fig. 9e), made

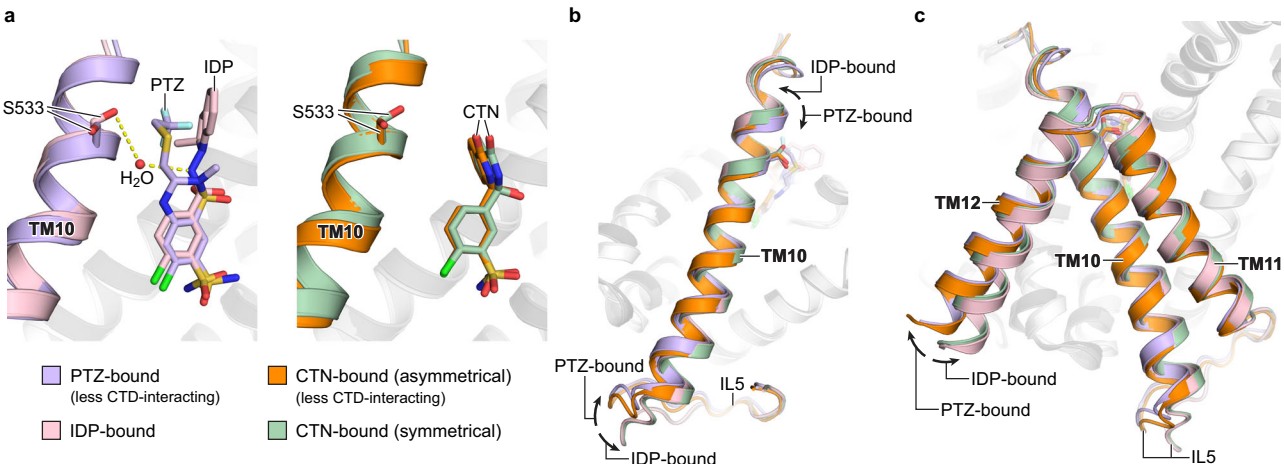

**Fig. 5 | Ligand-specific conformational changes. a** Ligand-specific interactions with S533. The PTZ-bound (less CTD-interacting), IDP-bound, asymmetrical CTN-bound (less CTD-interacting), and symmetrical CTN-bound NCC structures are colored purple, pink, orange, and green, respectively. Polar interactions are shown as yellow dashed lines. **b**, **c** Ligand-specific conformational changes of TM10-12. The less CTD-interacting TMD subunits of PTZ-bound and asymmetrical CTN-bound structures are aligned to the TMD subunits of IDP-bound and symmetrical CTN-bound structures. The color scheme is the same as in (**a**). TM11 and TM12 are not shown in **b** for clarity.

hNCC approximately 40-fold less sensitive to chlorthalidone-mediated inhibition while it reduced the sensitivities to indapamide- and polythiazide-mediated inhibition only by ~2-fold (Fig. 4b, c). This is consistent with chlorthalidone being closest to G230, and thus most sensitive to substitutions of G230.

### Impact of NCC polymorphisms on thiazide diuretic efficacy

The G230A variant's significant differential impact on NCC's sensitivity to thiazide diuretics suggests local structural/conformational changes of G230 are important for influencing thiazide diuretics' potencies. G230D (rs375990084) is the only reported genetic variation affecting G230 in the Single Nucleotide Polymorphism Database (dbSNP)[45], but it severely impairs hNCC transport function (Supplementary Fig. 9d), which is consistent with being a disease-causing mutation[46]. The two residues next to G230 show polymorphisms in the same database: V229M (rs147184383; allele frequency ~0.005%, and can be as high as 0.03–0.07% in certain populations, including Korean[47], Japanese[48], Dominican, native Hawaiian, and Central American[49]) and V231M (rs2055073548; allele frequency 0.0004%). These two variants might potentially cause local conformational changes around G230 as the sidechains of V229 and V231 are in range to contact surrounding residues. We tested whether these two common polymorphisms affect NCC's sensitivity to thiazide diuretics.

We found the hNCC(V229M) variant enhances transport activity without affecting surface expression in the mammalian cell-based system (Supplementary Fig. 9e, f). Since the side chain of V229 faces more closed and open local environments in the inward- and outward-facing conformations, respectively, it is possible that V229M substitution, with a larger side chain, might facilitate the inward-to-outward conformational transition to enhance hNCC transport activity (Supplementary Fig. 9i). Given the association between increased NCC activity and elevated blood pressure[50], this raises the possibility that carriers might have a higher risk of developing hypertension. Interestingly, while the V229M variation reduced hNCC's sensitivity to all three molecules, the effect was approximately half as severe for polythiazide (~5-fold reduction) compared to chlorthalidone and indapamide (~10-fold reductions) (Fig. 4d and Supplementary Fig. 9g). This difference might be due to the longer distance between the 3-position -CH$_2$-S-CH$_2$-CF$_3$ moiety of polythiazide and the G230 region, compared with the moieties on chlorthalidone and indapamide that are positioned near G230.

To further test this, we examined hydrochlorothiazide, a commonly used thiazide-type diuretic that lacks the bulky substituent (-CH$_2$-S-CH$_2$-CF$_3$) at the 3-position of the benzothiadiazine core that is found in polythiazide. This difference may provide extra space to alleviate steric hindrance caused by local conformational changes around G230 that are induced by the larger sidechain of V229M. Indeed, the V229M variant only reduced hNCC's sensitivity to hydrochlorothiazide by twofold (Fig. 4d and Supplementary Fig. 9g), substantially less severe than the tenfold reduction for chlorthalidone and indapamide.

Similar to the V229M polymorphism, the V231M polymorphism, which does not significantly impact hNCC transport activity (Supplementary Fig. 9e), also had differential effects on hNCC's sensitivity to the four molecules. The hNCC(V231M) variant is nearly as sensitive as the wild-type to polythiazide- or hydrochlorothiazide-mediated inhibition (Fig. 4d and Supplementary Fig. 9h). In contrast, hNCC(V231M) was less sensitive than the wild-type to chlorthalidone- and indapamide-mediated inhibition (3- and 2-fold lower, respectively) (Fig. 4d and Supplementary Fig. 9h). Thus, for hypertensive patients with the V231M polymorphism, hydrochlorothiazide might also be a better choice than chlorthalidone and indapamide.

## Discussion

Our structures of hNCC in complex with two widely used thiazide-like diuretics, chlorthalidone and indapamide, suggest they inhibit hNCC transport function through mechanisms that they share with thiazide-type diuretics (as exemplified by polythiazide). Both thiazide-like (chlorthalidone and indapamide) and thiazide-type (polythiazide) diuretics bind and stabilize hNCC in an outward-facing conformation (Fig. 6a). These three molecules share a binding site at the bottom of the extracellularly open vestibule in the TMD of hNCC, and they exhibit similar binding poses, which suggests that their shared benzene-sulfonamide moiety is a key driver for NCC binding. These observations support previous reports that all three molecules compete with metolazone, a benzenesulfonamide-containing thiazide-like diuretic, for NCC binding[39]. When bound to hNCC, these three molecules not only occupy the Cl$^-$-binding site through their chlorine groups; they also provide steric hindrance that impedes hNCC's conformational switch from an outward-facing to an inward-facing conformation (Fig. 6b, c). Thus, their inhibitory effect arises from competition with the substrate and interruption of the transport cycle. This mechanism

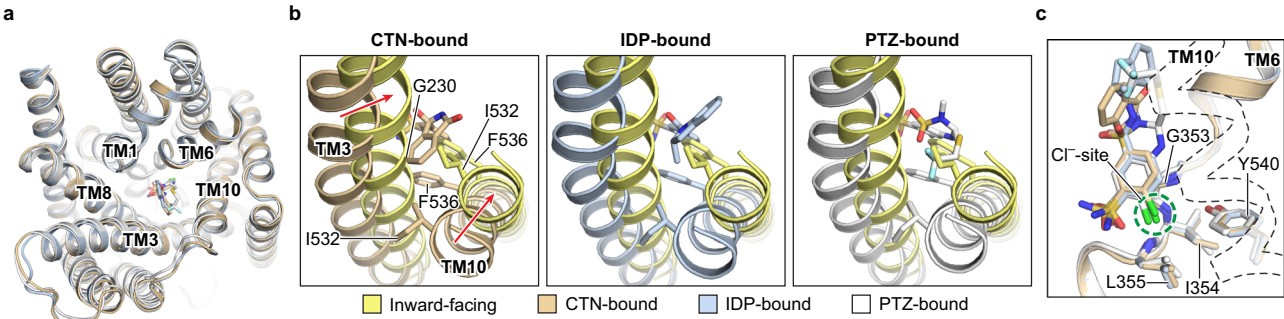

**Fig. 6 | NCC inhibition mechanisms of different thiazide diuretics.**
**a** Superposition of the TMDs of CTN-, IDP-, and PTZ-bound NCC structures, which are colored tan, light blue, and white, respectively. The color scheme is the same in (**b**, **c**). The bound ligands are shown as color matched sticks. For clarity, the cap domains are not shown. **b** Structural comparisons of TM3 and TM10 in the inward-facing (yellow) and outward-facing (CTN-, IDP-, or PTZ-bound) NCC. Movements of individual helices during outward-facing to inward-facing conformational switches are indicated with red arrows. **c** Relationships between NCC's Cl⁻-binding site and the chlorine groups of bound chlorthalidone, indapamide, and polythiazide molecules. The Cl⁻-binding site is outlined with a green dashed circle, and Cl⁻-coordinating residues are shown as sticks. TM10 is shown as a dashed outline for clarity.

is likely shared by other thiazide-like diuretics given that they, although structurally diverse, mostly are derivatives of benzenesulfonamide with a chlorine group ortho to the sulfamoyl group[24], which presumably can occupy a similar position in the substrate translocation cavity.

The shared NCC-inhibition mechanisms among thiazide-type and thiazide-like diuretics help explain the intriguing observation that eel βNCC is resistant to all tested thiazide-type and thiazide-like diuretics[51,52]. In the thiazide-binding pocket, four residues are different between hNCC (αNCC) and eel βNCC: the residues corresponding to hNCC G230, H234, T352, and C472 are A248, N252, I370, and F491, respectively, in eel βNCC. We have shown that G230A substitution only has minor effects on hNCC sensitivity to polythiazide- and indapamide-mediated inhibition. Both hNCC(H234N) and hNCC(T352I) variants could also be completely inhibited by 100 μM polythiazide (Supplementary Fig. 8j). In contrast, the transport activity of hNCC(C472F) variant was inhibited by only ~50% under the same condition (in the presence of 100 μM polythiazide) (Supplementary Fig. 8j). Therefore, C472F substitution severely impaired hNCC sensitivity to polythiazide-mediated inhibition, causing an approximately 200,000-fold decrease in potency (IC₅₀ shifted from ~0.5 nM to ~100 μM). Given that the side chain of C472 is close to all three thiazide diuretics in our structures, the bulky side chain introduced by C472F substitution likely results in steric clashes, thereby interfering with drug binding (Supplementary Fig. 8g–i). Based on these findings, F491 appears to be the main contributor to the thiazide resistance of eel βNCC.

Intriguingly, structures of hNCC in complex with polythiazide, chlorthalidone, and indapamide suggest differences in intradimer and interdomain dynamics of NCC. The indapamide-bound hNCC adopts a symmetric dimer structure with extensive interactions between TMDs and CTDs in both subunits, unlike the asymmetric dimer with a tilted CTD dimer observed in the polythiazide-bound structure. The distances between the two TMD subunits and the angles of relative rotation between the TMD and CTD dimers also differ between these structures. In the chlorthalidone-bound structures, both symmetrical and asymmetrical dimer configurations, resembling those seen in the indapamide- and polythiazide-bound structures, respectively, were observed. The presence of different dimer configurations, along with a less well-resolved TMD subunit in the asymmetrical dimer, suggests that hNCC potentially exhibits greater intradimer dynamics in the chlorthalidone-bound state. Additionally, the CTD-interacting region of the NTD appears to be more flexible in the indapamide- and chlorthalidone-bound states compared to the polythiazide-bound state. These apparent differences in domain-domain interaction modes might potentially be associated with the inhibitor binding.

Given these inhibitors vary in potency, they may differentially stabilize the bound conformation, impacting TMD conformational dynamics. Furthermore, they may cause subtle conformational differences in the TMD due to specific protein-ligand interactions (Fig. 5). These differences may, in turn, affect the relative stability of specific domain-domain interaction modes, as NTD-CTD-TMD interactions are interconnected and potentially conformation-dependent.

The greater intradimer dynamics in the chlorthalidone-bound state are consistent with a recent chlorthalidone-bound hNCC structure (PDB: 8VPP)[53], reported during the review process of this work, which resolved only one TMD (Supplementary Fig. 10a). Despite one unresolved TMD subunit, this structure shares similar overall architecture, chlorthalidone-binding poses, and protein-ligand interactions with our asymmetrical chlorthalidone-bound NCC_cryo structure (overall RMSD = 1.697 Å; TMD RMSD = 1.081 Å) (Supplementary Fig. 10a–c). Intriguingly, although the indapamide-bound hNCC structure (PDB: 8VPN) from the same study[53] shares similar indapamide-binding poses and protein-ligand interactions (TMD RMSD = 0.427 Å), it adopts a different dimer configuration compared to our indapamide-bound structure (Supplementary Fig. 10d–f). It exhibits an asymmetrical dimer with a tilted CTD and one less well-resolved TMD subunit, where only the well-resolved TMD subunit is outward-facing and ligand-bound, while the other adopts a thiazide-free inward-facing conformation. In contrast, our indapamide-bound NCC_cryo structure forms a symmetrical dimer with both TMD subunits outward-facing and ligand-bound. Moreover, the reported asymmetrical dimer configuration differs from the asymmetrical dimers observed in our polythiazide- and chlorthalidone-bound structures (Supplementary Fig. 10g). These differences in dimer configuration might potentially represent different states, as multiple dimer modes have been previously observed in NCC under the same ligand-bound conditions[36] and related CCC transporters[54–57]. Given that the E240A mutation lies far away from the TMD-TMD and TMD-CTD interfaces, the observed differences may potentially stem from differences in sample environment: while NCC_cryo samples were reconstituted in nanodiscs, hNCC structures were determined in detergent micelles. A lipid bilayer-like environment, compared to detergent micelles, may influence both TMD-TMD and TMD-CTD interactions, as lipids partially mediate TMD interactions and the CTD is positioned to contact either the lipid bilayer or detergent micelles.

The potential changes in interdomain and intradimer dynamics of hNCC in complex with different thiazide diuretics call for further biophysical studies, such as Förster resonance energy transfer (FRET)[58] and electron paramagnetic resonance (EPR) spectroscopic[59] studies. It also remains to be investigated whether and how the differences in

intradimer and interdomain dynamics of hNCC might impact properties other than hNCC's basic transport function, such as protein-protein interactions, intracellular distribution, and post-translational modification. Characterizing these aspects of hNCC-inhibitor interactions might provide important hints that illuminate the physiological and clinical implications of differing effects between thiazide-type and thiazide-like diuretics, such as potential better cardioprotective effects of thiazide-like diuretics[26].

Thiazide-type and thiazide-like diuretics are widely used and are frequently viewed as interchangeable, such that equipotent doses are expected to achieve similar levels of blood pressure reduction[19,25]. Yet, here, based on structural insights into inhibitor-specific interactions between thiazide diuretics and hNCC and structure-function analyses, we found that hNCC polymorphisms can impact sensitivities to various thiazide-type and thiazide-like diuretics to significantly differing degrees, due to differences in ligand-protein interactions. Particularly, we found that the V229M variant reduced hNCC's sensitivity to hydrochlorothiazide significantly less than to chlorthalidone and indapamide (2-fold vs. 10-fold). These observations carry important potential clinical implications: given thiazide diuretics have dose-dependent off-target metabolic side effects, such as impaired glucose tolerance[60–62] and hyperuricemia[63,64], their doses cannot be increased arbitrarily to overcome resistant NCC polymorphisms. For example, a chlorthalidone dose just 2- to 4-fold over the currently recommended level[65] caused elevated levels of serum glucose and uric acid[66]. Therefore, it would be impractical to treat hypertensive patients with the V229M polymorphism using chlorthalidone at 10-fold the recommended dose. In such cases, hydrochlorothiazide would likely be a better choice than the other three drugs given its much lower sensitivity to the V229M polymorphism. In addition to the studied V229M and V231M, there are other polymorphisms affecting thiazide-interacting residues, which potentially may have differential effects on hNCC sensitivity to different thiazide diuretics (Supplementary Table 2).

Given that current practice aims to use low-dose thiazide-type and thiazide-like diuretics to treat hypertension while avoiding metabolic side effects[67,68], patients harboring NCC polymorphisms with differential sensitivities to specific thiazide diuretics require properly chosen thiazide diuretics to achieve adequate efficacy while minimizing metabolic side effects. This highlights the importance of NCC polymorphism in pharmacogenomics of thiazide diuretic therapy. Our structural insights into NCC inhibition by various thiazide diuretics can thus provide a molecular blueprint to facilitate the identification and interpretation of pharmacologically significant polymorphisms.

## Methods

### Expression and purification

The NCC$_{cryo}$ construct consists of human NCC with its NTD (first 131 residues) replaced by that of zebrafish NKCC1 (first 202 residues), along with an E240A mutation and a *Strep*-tag II[69] at the N-terminus. The codon-optimized NCC$_{cryo}$ was cloned into a modified BacMam vector[70], which was then used for generating baculoviruses. HEK293S cells ($3.0–3.5 \times 10^6$ cells per mL) were infected with the baculoviruses for NCC$_{cryo}$, with sodium butyrate (10 mM) added 12–15 h afterwards, and the incubation temperature reduced to 30 °C. Two days after the addition of sodium butyrate, HEK293S cells were harvested.

To purify NCC$_{cryo}$ in complex with chlorthalidone, HEK293S cells were homogenized in a pre-cooled low-chloride buffer (20 mM HEPES pH 7.4 (titrated with KOH), 50 mM Na$_2$SO$_4$, 25 mM K$_2$SO$_4$, and 1.5 mM KCl) containing 25 μM chlorthalidone (Thermo Scientific Chemicals) and protease inhibitors using a Dounce homogenizer. The cell lysate was subjected to centrifugation ($40,000 \times g$, 40 min) to prepare the crude membrane. The homogenized crude membrane (in the low-chloride buffer supplemented with 250 μM chlorthalidone and protease inhibitors) was subjected to solubilization for 1 h in 1% lauryl

maltose neopentyl glycol (LMNG; Anatrace) supplemented with 0.2% cholesteryl hemisuccinate (CHS; Anatrace) at 4 °C. The insoluble fraction was pelleted by centrifugation ($40,000 \times g$, 40 min). The Strep-Tactin Sepharose resin (IBA) was mixed with the supernatant and incubated at 4 °C for 1 h. Washing was performed using the low-chloride buffer with 0.01% LMNG, 0.002% CHS, and 250 μM chlorthalidone. Afterwards, bound proteins were eluted with 10 mM desthiobiotin in the wash buffer. Further size-exclusion chromatography purification was carried out via Superose 6 increase column (Cytiva) in the low-chloride buffer supplemented with 0.003% LMNG, 0.0006% CHS, and 500 μM chlorthalidone. Purified NCC$_{cryo}$ proteins in the peak fractions were collected for nanodisc reconstitution. NCC$_{cryo}$ in complex with indapamide was purified using the same method except replacing chlorthalidone with indapamide (Sigma-Aldrich).

### Nanodisc reconstitution

To prepare the nanodisc sample of NCC$_{cryo}$ in complex with chlorthalidone, purified NCC$_{cryo}$ proteins (in the low-chloride buffer supplemented with 0.003% LMNG, 0.0006% CHS, and 500 μM chlorthalidone), MSP1E3D1 proteins, and lipids were mixed (1:4:240). The MSP1E3D1 proteins were purified in the low-chloride buffer. The lipid sample used was a mixture of DOPE, POPS, and POPC (Avanti) in a molar ratio of 2:1:1. Individual lipids were prepared by solubilizing with the low-chloride buffer containing 27 mM sodium cholate (Sigma-Aldrich). Extra sodium cholate was added to the mixture of NCC$_{cryo}$, MSP1E3D1, and lipids to make the final concertation of sodium cholate 20 mM. The mixture of NCC$_{cryo}$, MSP1E3D1, and lipids was rotated (4 °C, 1 h) to allow complete mixing. Two batches of 0.125 g (dump weight) Bio-Beads (Bio-Rad) equilibrated with the low-chloride buffer were added (separated by 30 min) to remove detergents. After overnight incubation, Bio-Beads were removed using Spin-X centrifuge tube filters (Corning). The size-exclusion chromatography (Superose 6 increase column) purification was carried out on the reconstituted sample in the low-chloride buffer containing 500 μM chlorthalidone. Peak fractions containing nanodisc reconstituted NCC$_{cryo}$ proteins were concentrated (~8 mg/mL) for cryo-EM. The procedures of reconstituting NCC$_{cryo}$ in complex with indapamide into MSP1E3D1 nanodiscs were the same as those for the chlorthalidone sample, except replacing chlorthalidone with indapamide. The final concentration of indapamide-bound NCC$_{cryo}$ in nanodisc used for cryo-EM sample preparation was ~8.5 mg/mL.

### $^{125}$I⁻-uptake assay

HEK293S cell lines, cultured in 293 Freestyle medium (Life Technologies; plus 10% fetal bovine serum), stably expressing hNCC and its variants were generated following published protocols[71]. Cells were transfected using Lipofectamine 3000 (Invitrogen) with a pSBbi-RP-based plasmid containing hNCC or its variants and the SB100X vector. Puromycin (2 μg/mL; Gemini Bio-Products) was used for selection of stable cell lines.

To prepare for the uptake experiment, cells were plated at $0.5 \times 10^6$ cells/well in 24-well plates one day in advance. To activate NCC, cells were treated with a chloride-free hypotonic solution composed of 67.5 mM sodium-, 2.5 mM potassium-, 0.25 mM magnesium-, and 0.25 mM calcium-gluconate, as well as 7.5 mM HEPES pH 7.4 (buffered using Tris) at 37 °C for 10 min[72]. After the activation step, the experiments were carried out at room temperature. Ten minutes before the start of $^{125}$I⁻ uptake, 70 μM Bumetanide (Thermo Scientific Chemicals) and 100 μM 4,4′-diisothiocyanatostilbene-2,2′-disulfonic acid (DIDS; Sigma) were added to suppress background iodide uptake activity. Indicated thiazide diuretics were also added to the indicated concentrations 10 min before the start of $^{125}$I⁻ uptake. The original buffer was switched to an uptake buffer composed of 135 mM Na-gluconate, 5 mM K-gluconate, 1 mM Mg-gluconate, 1 mM Ca-gluconate, 70 μM bumetanide, 100 μM DIDS, 1 mM NaI, 0.25 μCi/ml Na$^{125}$I (American

Radiolabeled Chemicals, Inc.), and 15 mM HEPES pH 7.4 (buffered using Tris), in the presence or absence of thiazide diuretics to initiate $^{125}I^-$ uptake. The $^{125}I^-$ uptake process lasted 1 h and was stopped by two rapid washes using ice-cold 110 mM MgCl$_2$. Cells lysed using 1% sodium dodecyl sulfate (SDS) were analyzed by scintillation counting.

To calculate the fold change of the half-maximal inhibitory concentration (IC$_{50}$) between hNCC(WT) and different hNCC variants, individual dose-response curves were normalized first using the smallest mean and the value of the lowest concentration as 0% and 100%, respectively. The normalized data were then fitted using the "EC50 shift" nonlinear regression in GraphPad Prism 10 with the shared EC50Control assigned to be the IC$_{50}$ of hNCC(WT). The following constraints were used in the EC50 shift analysis. The EC50Control was set to be shared across all curves under analysis and constrained to be greater than 0. The EC50Ratio was constrained to be greater than 0. The bottom, top, and HillSlope were constrained to be 0, 100, and −1, respectively. For Fig. 1f, g and Supplementary Fig. 2d, GraphPad Prism 10 was used to perform curve fitting and IC$_{50}$ calculation for normalized data using $y = 100/(1 + (x/IC_{50}))$.

### Cell-surface expression

A previously developed surface staining method was used to measure the surface expression levels of hNCC variants[36]. Briefly, a haemagglutinin (HA) epitope was inserted between residues 414 and 415 of hNCC variants, which enabled labeling of surface-expressing tagged proteins given its location in the extracellular loop 4. The HA-tagged hNCC variants were cotransfected with a yellow fluorescent protein (YFP), which was used as an indicator of successful transfection, into HEK293S cells. After washing with phosphate-buffered saline (PBS), $0.5 \times 10^6$ cells were incubated with anti-HA antibody (Alexa Fluor 647-conjugated; BioLegend #682404; 4 µg/ml). After 30 min, cells underwent three PBS washes before subjecting to flow cytometry analysis (BD Accuri C6 Plus). FITC-positive YFP-expressing cells were gated for surface expression quantification by measuring Alexa 647 mean fluorescence intensity (Supplementary Fig. 11).

### Cryo-EM sample preparation and data acquisition

The cryo-EM samples of the nanodisc reconstituted NCC$_{cryo}$ in complex with chlorthalidone or indapamide were prepared on holey carbon grids (Quantifoil R 1.2/1.3 and Au 300 mesh), using Vitrobot Mark IV (Thermo Fisher Scientific). Glow-discharged grids with 3 µl protein samples were blotted using the following parameters: 5 s pre-blotting time, 3 s blotting time, and level 3 blotting force (with humidity set to 100% and temperature set to 4 °C), then plunge-frozen in liquid ethane.

The cryo-EM data were acquired using a 300 kV Titan Krios microscope (with a GIF Quantum energy filter and K3 Summit detector) in the super-resolution mode (0.86 Å physical pixel size). Thermo Fisher Scientific's EPU software was used for data collection (40 movie frames over a 1.9 s exposure, 50 electrons/Å$^2$ total dose, and -1.2 to -2.0 µm defocus).

### Cryo-EM data processing

CryoSPARC Live v4.6.2[73] was used to process 5199 movies of the indapamide-bound NCC$_{cryo}$ data from motion correction and Patch CTF estimation to particle picking and extraction using a box size of 80 pixels in $4 \times 4$ binning. This resulted in 2,827,013 particles, from which a subset of 65,509 particles was identified through six 2D classification steps. Meanwhile, Topaz picking (v0.2.4)[74], which was trained on 48,623 particles curated from 2500 images, was carried out. This resulted in 1,988,788 particles (box size of 80 pixels, $4 \times 4$ binning). Three rounds of heterogeneous refinement were performed, following a previously published protocol[75], with reference maps (accurate and biased) from the chlorthalidone-bound NCC$_{cryo}$ dataset. The duplicates were removed. The non-uniform refinement on the remaining 446,219 particles (re-extracted; 320 pixels box size) produced a

reconstruction at 2.79 Å resolution. BlocRes[76] in cryoSPARC was used for assessing local resolutions.

Similarly, cryoSPARC Live v4.6.2 was used to process 5010 movies of the chlorthalidone-bound NCC$_{cryo}$ dataset. From the resulting 2,420,915 particles, 59,060 particles were chosen for downstream analyses following four rounds of 2D classification. At the same time, Topaz picking, trained on 17,940 particles from 2D classes representing different orientations, resulted in 1,706,941 particles (box size of 80 pixels, $4 \times 4$ binning). After two rounds of 3D classification guided by seeds[75] and the removal of duplicates, 463,404 particles were re-extracted (320 pixels box size). The heterogeneous refinement resulted in two distinct classes with 103,392 and 96,877 particles, respectively. The non-uniform refinement on these two groups of particles produced a reconstruction at 3.01 Å resolution with C2 symmetry and a reconstruction at 3.24 Å resolution with C1 symmetry, respectively. To better resolve the TM regions, 3D variability analysis[77] was carried out on the class with 96,877 particles, and the subsequent non-uniform refinement resulted in a reconstruction at 3.52 Å resolution.

### Model building and refinement

For the asymmetrical chlorthalidone-bound NCC$_{cryo}$, the structure of polythiazide-bound hNCC (PDB: 8FHN) was chosen as the initial template due to their similar symmetrical TMD dimer organizations. Given the better ligand resolution, the cryo-EM map without 3D variability analysis was used for model building. For both symmetrical chlorthalidone- and indapamide-bound NCC$_{cryo}$, one protomer (chain A) from the polythiazide-bound hNCC structure (PDB: 8FHO) served as the initial model. The 3D models of chlorthalidone and indapamide were generated using the LigPrep tool of the Schrödinger software suite[78] using the corresponding SMILES strings found in the PubChem database[79]. The SMILES strings used for chlorthalidone and indapamide were C1=CC=C2C(=C1)C(=O)NC2(C3=CC(=C(C=C3)Cl)S(=O)(=O)N)O and CC1CC2=CC=CC=C2N1NC(=O)C3=CC(=C(C=C3)Cl)S(=O)(=O)N, respectively. The LigPrep-generated PDB files of chlorthalidone and indapamide were used to generate their geometry restraints using the eLBOW (electronic Ligand Builder and Optimization Workbench) program module[80] of the Phenix suite[81]. The 3D model of ATP was generated using the "Get monomer" function in Coot[82], and the generated PDB file was used to generate geometry restraints using eLBOW. The symmetrical dimer structure of NCC$_{cryo}$ in complex with chlorthalidone or indapamide was then generated based on C2 symmetry. Coot was used for model building, and the models were refined using the Phenix suite with the geometry evaluated by MolProbity. PyMol (Schrödinger)[83] and UCSF ChimeraX[84] were used for figure generation.

### Reporting summary

Further information on research design is available in the Nature Portfolio Reporting Summary linked to this article.

### Data availability

The symmetrical and asymmetrical chlorthalidone-bound NCC$_{cryo}$ maps and the indapamide-bound NCC$_{cryo}$ map have been deposited in the Electron Microscopy Data Bank under the accession codes EMD-71664, EMD-71665, EMD-71666, with the corresponding models in the Protein Data Bank with accession codes 9PIE, 9PIF, 9PIG. Accession codes used in this study that have been previously published are 8FHN, 8FHO, 8FHT, 8VPP, 8VPN, EMD-29096 (polythiazide-bound map used in Supplementary Fig. 5a, d), and EMD-29097 (polythiazide-bound map used in Supplementary Figs. 8a and 9a, b). Source Data are provided with this paper as a Source Data file.

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

## Acknowledgements

We thank L. Montabana and M. Zaoralová at Stanford cEMc for help with EM data collection. Some of this work was performed at the Stanford-SLAC Cryo-EM Center (S2C2), which is supported by the National Institutes of Health Common Fund Transformative High-Resolution Cryo-Electron Microscopy program (U24 GM129541). The authors would also like to thank the following S2C2 personnel for their invaluable support and assistance: Dr. Yan Liu. This work was made possible by support from Stanford University, NIH R35GM153424, and the Harold and Leila Y. Mathers Charitable Foundation (L.F.).

## Author contributions

C.-L.L. and J.Z. carried out experimental work. L.F. directed the project. C.-L.L. and L.F. wrote the manuscript with input from J.Z.

## Competing interests

The authors declare no competing interests.
