## [Transparent Peer Review file · Nature Communications]

Molecular mechanisms of thiazide-like diuretics-mediated inhibition of the human Na-Cl cotransporter

Corresponding Author: Dr Liang Feng

Version 0:

Reviewer comments:

Reviewer #1

(Remarks to the Author)

In this work, the authors carry out structural and functional studies to investigate the mechanisms of NCC inhibition mediated by the thiazide-like diuretics indapamide and chlorthalidone.

The same group recently reported the cryo-EM structure of NCC bound to a thiazide, polythiazide; This study provided important insights into the mechanisms of NCC inhibition; however, questions remain about the processes involved in its inhibition.

To obtain cryo-EM maps, the authors started from an NCC construction in which the amino-terminal region was replaced by the amino-terminal region of the zebrafish NKCC1 (DrNKCC1), and some mutations such as E240A, S345E, and L525C, were also incorporated to achieve the open conformation of the NCC towards the extracellular space, they called this modified NCC as NCCcryo.

Major concerns

I understand that all the functional experiments were done in cells transfected with hNCC, not with the NCCcryo construct in which the amino-terminal domain was substituted with DrNKCC1 and containing the E240A, S345E, and L525C substitutions. I believe that it is required that the authors show to what extent these modifications in NCC affect its function to evaluate the effect on thiazide inhibition properties of the different mutations presented.

Also, can the authors present the comparison of Cryo-EM of the wild type NCC in comparison with the NCCcryo to show to what extent the conformation of the transporter was changed?

In supplementary figure 4, the final map of NCCcryo-chlorthalidone lacks the TM regions of a monomer, mentioning that the intra-dimeric dynamics are greater in this complex. Because this is not observed in the maps of cryo-EM NCC-Indapamide and NCC-polythiazide, could there be another explanation for this? Could it be because the monomer that is not seen has another structure and another state of opening towards the extracellular space?

Could the authors provide a further explanation for the absence of this segment in the final NCCcryo-chlorthalidone maps?

The authors show that the residues N149 and N227 are critical for the chlorthalidone, polythiazide, and indapamide inhibition of NCC since the substitution of these residues with alanine decreased the IC50 for these diuretics. However, in previous work (Sci. Adv. 8 (45), eadd7176. 2022) it was shown that these substitutions result in loss of the NCC function. Thus, the authors must show the I125 uptake in wild type and N149A,N227A mutant NCC to demonstrate to what extent the NCC function is preserved.

In this regard, Moreno et al. (JBC, vol.291, No.43. 2016 and AJP Cell Physiol 323: C385-C399, 2022) showed that NCC from eel is a NaCl cotransporter that is not sensitive to thiazides, but most of the residues suggested in this work are present in this orthologue. Could the authors revise the eel NCC sequence and make comments about the potential explanations for not being sensitive to thiazides, although the residues are conserved?

Also, residues like H234, T352, and C472 that are suggested to be critical for the Na-Cl transport are not present in eel NCC sequence which however perform as Na-Cl cotransporter.

Regarding the single nucleotide polymorphisms in NCC, in addition to V229M and V231M, could the authors provide a table showing SNPs located in the critical thiazide and thiazide-like binding sites and comment on potential effects?

Reviewer #2

(Remarks to the Author)

The manuscript investigates how thiazide-like diuretics, chlorthalidone and indapamide, inhibit the human Na-Cl cotransporter (hNCC). By combining structural and functional studies, the research identifies both commonalities and differences in inhibition mechanisms between these and thiazide-type diuretics. The structural analysis uncovers genetic variations in hNCC that influence diuretic potency, offering insights into NCC's molecular pharmacology and suggesting avenues for precision medicine in hypertension management. Overall, this manuscript is well prepared. However, some revisions are needed:

1. Comparing Supplementary Fig. 3f and Fig. 4g, it appears that Fig. 4g lacks some angular distributions, which might affect the density quality in the chlorthalidone-bound hNCC structure. Also, the TMD is missing in the final map, given the small data set, could additional data collection and 3D variability analysis address this issue?
2. In analyzing the indapamide-binding site, water-mediated interactions are crucial. Binding assays should include T352A to assess how substituting alanine affects hNCC sensitivity to indapamide inhibition.
3. The Single Nucleotide Polymorphism Database identifies G230D as a disease-causing mutation. Since the article only covers G230A, including data on G230D would provide a more comprehensive analysis.
4. Although hNCC with indapamide and chlorthalidone shows a similar overall structure to the polythiazide-bound form, differences in the distance between TMD subunits are evident. It is essential to label the distance between the TMDs in Fig1b, and measure the RMSD to understand the underlying causes of the variations between the two drugs.

Reviewer #3

(Remarks to the Author)

The manuscript by Lee and colleagues presents two cryo-EM structures of an engineered human NCC transporter in complex with thiazide-like diuretics, specifically chlorthalidone and indapamide. This work is a "sequel" to the authors' previous work on the hNCC transporter¹, offering valuable new insights into the mechanisms of NCC inhibition by thiazide-like and thiazide-type diuretics. The structural and functional data suggest that the shared benzenesulfonamide moieties of these diuretics interact with NCC in similar ways, while the distinct functional groups of each diuretic form unique interactions with the transporter. A notable aspect of this study is the demonstration that polymorphisms could influence NCC's response to different diuretics. The mechanisms uncovered in this work are significant and could contribute to the refinement of existing diuretics for hypertension treatment. While the manuscript is undoubtedly of high importance to the field, I have several concerns and suggestions that should be addressed.

-- It is surprising to see that only one TMD is resolved in chlorthalidone-bound hNCC structure. Given similar binding affinity between indapamide (IC₅₀=9.3 nM) and chlorthalidone (IC₅₀=24 nM) to the hNCC, what could account for the markedly different intradimer dynamics caused by their binding to the transporter?

--Based on the 2D classification results shown in Supplementary Figure 4c, it appears that both TMDs exhibit clear features in some classes. Did the author ever isolate any particles that results in a map with an intact TMD dimer? Please show some reference maps used in 3D classification. Also, why were the particles used for the final map reconstruction (Supplementary Figure 4a) significantly fewer than those in the indapamide-bound dataset?

--The FSC curves in Supplementary Figure 3e do not converge to 0.

--For better comparison, please highlight the NTD in both maps and models shown in Figures 1a and 1b.

--There is no mention of ATP/ADP in the entire manuscript, which is somewhat surprising, given that adenine nucleotide binding was reported to be important in the previous study¹. Are there any ATP/ADP densities present in either of the maps?

--Line 191. "hydrophobic interactions with G230, M233, I532, and S533". A figure call is suggested here, as G230 is not depicted until Figure 4a.

--Line 205. "For C472, although its side chain interacts with all three molecules, it appears to make the strongest interactions with polythiazide." What is the supporting evidence?

--What is the transport activity of the G230A mutant? Additionally, I am curious about how the G230D variant affects transport activity; did the authors test this?

--Can the authors provide insight into why the V229M variant exhibits enhanced transport activity from a structural and mechanistic perspective?

--To better understand the effects of diuretics on different mutants, please provide the IC50 values from the dose-response curves.

--Why were different detergents used during the SEC step? For the chlorthalidone-bound sample, detergent exchange to GDN was performed, whereas LMNG/CHS was used throughout the preparation for the indapamide-bound sample. Please provide a rationale for this choice.

--Regarding cryo-EM sample preparation, 3 µl of the protein sample was applied to the grids. Are any surfactants required during this step?

Reference:

1. Fan, M., Zhang, J., Lee, C.-L., Zhang, J. & Feng, L. Structure and thiazide inhibition mechanism of the human Na–Cl cotransporter. *Nature* 1–6 (2023) doi:10.1038/s41586-023-05718-0.

Version 1:

Reviewer comments:

Reviewer #1

(Remarks to the Author)

I agree with all changes made by the authors.

In the original description of the polythiazide binding site the G320 was not included and the A529 was included. But not, G320 was mentioned as part of the site, while A529 is not. Please clarify.

Reviewer #2

(Remarks to the Author)

Thank you for the excellent revisions. All my questions have been addressed, and I have no further comments.

Reviewer #3

(Remarks to the Author)

The manuscript has undergone significant improvement, and all of my concerns have been addressed. Congratulations to the authors for their exceptional work.

Open Access This Peer Review File is licensed under a Creative Commons Attribution 4.0 International License, which permits use, sharing, adaptation, distribution and reproduction in any medium or format, as long as you give appropriate credit to the original author(s) and the source, provide a link to the Creative Commons license, and indicate if changes were

made.

We thank reviewers for their insightful and constructive comments. We have followed their advice and thoroughly addressed all of their points, which we believe has substantially improved the manuscript. Below, we include the referees' comments, followed by our responses.

REVIEWER COMMENTS

Reviewer #1 (Remarks to the Author):

In this work, the authors carry out structural and functional studies to investigate the mechanisms of NCC inhibition mediated by the thiazide-like diuretics indapamide and chlorthalidone.

The same group recently reported the cryo-EM structure of NCC bound to a thiazide, polythiazide; This study provided important insights into the mechanisms of NCC inhibition; however, questions remain about the processes involved in its inhibition. To obtain cryo-EM maps, the authors started from an NCC construct in which the amino-terminal region was replaced by the amino-terminal region of the zebrafish NKCC1 (DrNKCC1), and some mutations such as E240A, S345E, and L525C, were also incorporated to achieve the open conformation of the NCC towards the extracellular space, they called this modified NCC as NCC_{cryo}.

We appreciate the reviewer's constructive suggestions. We have followed the reviewer's advice to address specific comments. We believe that these revisions have significantly strengthened the manuscript.

Major concerns

- 1. I understand that all the functional experiments were done in cells transfected with hNCC, not with the NCC_{cryo} construct in which the amino-terminal domain was substituted with DrNKCC1 and containing the E240A, S345E, and L525C substitutions. I believe that it is required that the authors show to what extent these modifications in NCC affect its function to evaluate the effect on thiazide inhibition properties of the different mutations presented.*

Response:

We appreciate the reviewer's comment on the functional effects of introduced mutations for structural study. Our construct used for cryo-EM study (NCC_{cryo})

contains NTD from DrNKCC1 and two mutations (S345E and L525C). In the following discussion, we will call it $\text{NCC}_{\text{chimera}}(\text{S345E/L525C})$ to distinguish from other constructs. In our radioactive iodide uptake assay, $\text{NCC}_{\text{chimera}}(\text{S345E/L525C})$ did not have detectable thiazide-sensitive iodide transport activity (Response Fig. 1a), presumably because S345E and L525C double mutations lock NCC in an outward-facing conformation. Although these two mutations are away from the thiazide binding pocket, suggesting they are unlikely to affect thiazide binding, to address the concern about the loss of function caused by these double mutations, we turned to $\text{NCC}_{\text{chimera}}(\text{E240A})$, which is functional (Response Fig. 1a) and has very similar sensitivities to chlorthalidone and indapamide as the hNCC(WT) (Response Fig. 1b).

We therefore used $\text{NCC}_{\text{chimera}}(\text{E240A})$ to capture the chlorthalidone- and indapamide-bound structures. Initially, $\text{NCC}_{\text{chimera}}(\text{E240A})$ in detergent micelles did not yield high-resolution cryo-EM structures of the chlorthalidone- and indapamide-bound forms. Consequently, we reconstituted $\text{NCC}_{\text{chimera}}(\text{E240A})$ into nanodiscs, which provide a bilayer-like environment more similar to the plasma membrane, the native site where NCC binds thiazide-like diuretics. With $\text{NCC}_{\text{chimera}}(\text{E240A})$ in nanodisc, we determined both chlorthalidone- and indapamide-bound structures at 3.01 Å and 2.79 Å resolution, respectively. These structures showed effectively the same binding poses of chlorthalidone and indapamide as observed in $\text{NCC}_{\text{chimera}}(\text{S345E/L525C})$. Given $\text{NCC}_{\text{chimera}}(\text{E240A})$ is functional, with comparable sensitivities to chlorthalidone and indapamide as the hNCC(WT), we decided to focus on these new structures in the manuscript. We also included the functional data of $\text{NCC}_{\text{chimera}}(\text{E240A})$ in the manuscript (Fig. 1f,g and Supplementary Fig. 2c; page 3, line 73-75).

Response Fig. 1 Iodide uptake activity of different NCC constructs (a) and the sensitivity of $\text{NCC}_{\text{chimera}}(\text{E240A})$ to chlorthalidone and indapamide (b). (a) The values are normalized to that of hNCC(WT) in the absence of metolazone (MTZ). Data are

shown as mean \pm s.d. (n = 4 independent experiments). **(b)** Comparisons between hNCC(WT) and NCC_{chimera}(E240A) in sensitivities to chlorthalidone and indapamide. Data are shown as mean \pm SEM (n = 3 independent experiments).

2. *Also, can the authors present the comparison of Cryo-EM of the wild type NCC in comparison with the NCC_{cryo} to show to what extent the conformation of the transporter was changed?*

Response:

Since we have now focused on NCC_{chimera}(E240A), we carried out the comparison between this construct and WT. For chlorthalidone-bound structures, NCC_{chimera}(E240A) and hNCC(WT) (recently released PDB: 8VPP (PMID: 39143061)) share similar overall structures (Response Fig. 2a), with one TMD subunit less well resolved. For the well resolved TMD subunit, NCC_{chimera}(E240A) and hNCC(WT) adopt nearly identical structures (RMSD 1.081 Å) (Response Fig. 2b). The bound chlorthalidone molecules and interacting residues also adopt highly similar binding poses and orientations, respectively (Response Fig. 2c). We observed slight differences in the relative orientation between the TMD and the C-terminal domain (CTD). When aligned by one TMD subunit, there is a small degree of rigid-body rotation between the CTDs of NCC_{chimera}(E240A) and hNCC(WT).

Under indapamide-bound conditions, the TMD subunits of NCC_{chimera}(E240A) and hNCC(WT) (recently released PDB: 8VPN (PMID: 39143061)) that are bound with indapamide show nearly identical structures (RMSD 0.427 Å) (Response Fig. 2e). Interestingly, different dimer configurations were observed (Response Fig. 2d). Indapamide-bound NCC_{chimera}(E240A) (in nanodiscs) adopts a symmetrical dimer structure with both TMD subunits bound with an indapamide molecule in an outward-facing conformation and forming extensive interactions with the CTD. In contrast, indapamide-bound hNCC(WT) (in detergent micelles) adopts an asymmetrical dimer structure with one TMD subunit bound with an indapamide molecule in an outward-facing conformation and another TMD subunit in an inward-facing apo state. Despite the difference in dimer configuration, TMD subunits bound with indapamide maintain nearly identical structures with the bound indapamide molecules and interacting residues also in very similar binding poses and orientations, respectively (Response Fig. 2f).

Therefore, the construct choice shows negligible effect on TMD and thiazide binding. The differences in dimer configuration between indapamide-bound

NCC_{chimera}(E240A) and hNCC(WT) structures might potentially represent different states as multiple different modes have been previously observed in NCC (PMID: 36792826) and related CCC transporters (PMID: 35585053; PMID: 36306358; PMID: 34031912; PMID: 33310850). As the mutations are far away from the TMD-TMD and TMD-CTD interfaces, we suspect that the sample environment might contribute to such differences: while our NCC_{chimera}(E240A) sample was in nanodiscs, the hNCC(WT) sample was in detergent micelles. A lipid bilayer-like environment versus detergent micelles may influence TMD-TMD interactions, as lipids (partially) mediate TMD interactions, and TMD-CTD interactions, since the CTD would be in contact with either the lipid bilayer or detergent micelles. Nonetheless, further studies, such as those using the Förster resonance energy transfer (FRET) technique, are needed to probe the exact dimer configuration and dynamics in the indapamide-bound state. We have now included comparisons with hNCC(WT) in the manuscript (page 11-12, line 374-397 and Supplementary Fig. 10).

Response Fig. 2 Comparisons between thiazide-like diuretic-bound structures of NCC_{chimera}(E240A) and hNCC(WT). CTN: chlorthalidone; IDP: indapamide. (a-c) Structural comparisons of CTN-bound NCC_{chimera}(E240A) (wheat) and hNCC(WT) (blue). (d-f) Structural comparisons of IDP-bound NCC_{chimera}(E240A) (light blue) and hNCC(WT) (pink).

3. In supplementary figure 4, the final map of NCC_{cryo}-chlorthalidone lacks the TM regions of a monomer, mentioning that the intra-dimeric dynamics are greater in

this complex. Because this is not observed in the maps of cryo-EM NCC-Indapamide and NCC-polythiazide, could there be another explanation for this? Could it be because the monomer that is not seen has another structure and another state of opening towards the extracellular space?

Could the authors provide a further explanation for the absence of this segment in the final NCCcryo-chlorthalidone maps?

Response:

In the unsharpened map of chlorthalidone-bound NCC_{chimera}(S345E/L525C), most TM helices of the less well-resolved TMD have visible densities at a lower contour level (Response Fig. 3a). It clearly shows this TMD also adopts an outward-facing conformation similar to the better resolved TMD subunit. We also observed similar non-protein density (presumably chlorthalidone) at the bottom of the outward-facing vestibule, albeit at lower resolution (Response Fig. 3a). Therefore, both TMDs represent a ligand-bound, outward-facing state. The DeepEMhancer sharpening improves the details of the well-resolved TMD, but the less well-resolved TMD becomes even weaker with only the TM helices close to the dimer interface resolved (Response Fig. 3b).

Response Fig. 3 Unsharpened (**a**) and DeepEMhancer sharpened (**b**) cryo-EM maps of chlorthalidone-bound NCC_{chimera}(S345E/L525C). The bulky non-protein densities corresponding to chlorthalidone are colored in red.

For NCC_{chimera}(E240A), by collecting a larger data set, we were able to resolve two dimer configurations at high resolution from the chlorthalidone-bound sample. One structure adopts a symmetrical dimer configuration with widely separated TMD subunits, which is similar to the indapamide-bound structure. Another structure has an asymmetrical dimer configuration with the TMD subunits in contact with each other, which is similar to the polythiazide-bound structure. This raises the possibility

that chlorthalidone-bound NCC_{chimera}(S345E/L525C) may also potentially adopt these two configurations. However, the significantly smaller cryo-EM dataset may potentially not be sufficient to allow effective separation of the two configurations, which may result in one of TMD much less well resolved (i.e. one TMD well aligned but the other not well aligned). We have now focused our discussion on NCC_{chimera}(E240A) given the higher quality of maps and its functional relevance.

Unlike the chlorthalidone-bound samples, indapamide-bound structures only show one conformation: a symmetrical dimer with widely separated TMD subunits. In addition, all previously determined polythiazide-bound NCC structures adopt asymmetrical dimer configurations. Therefore, NCC seems to adopt ligand-dependent dimer configurations. Structural comparisons suggest that ligand-dependent protein-ligand interactions (particularly involving S533 and the extracellular end of TM10) are potentially the underlying mechanisms (Response Fig. 4). The bulky substituent of polythiazide is closer to TM10 compared to indapamide and chlorthalidone. When bound to polythiazide, the side chain of S533 points away from the central vestibule to accommodate the 3-position bulky substituent; whereas with indapamide, S533 points toward the vestibule, stabilized by a water-mediated interaction (Response Fig. 4a). These differential interactions between S533, along with the extracellular end of TM10, and polythiazide or indapamide presumably would induce conformational differences in TM10. Because the intracellular loop 5 (IL5) and TM11/12 following TM10 are involved in TMD-CTD interactions and the dimer interface, respectively, conformational differences in TM10 may contribute to the different NCC dimer configurations (Response Fig. 4b and c). For chlorthalidone, the bulky substituent is not as close to TM10 as polythiazide, and it does not interact with and stabilize S533 like indapamide (Response Fig. 4a). Furthermore, chlorthalidone is shorter than polythiazide and indapamide, and does not extend extracellularly beyond the level of S533 (Response Fig. 4a). Therefore, chlorthalidone potentially has a “neutral” effect on S533 and the extracellular end of TM10, and the side chain of S533 can potentially adopt different orientations in chlorthalidone-bound structures. In the symmetrical chlorthalidone-bound structure, S533 points toward the vestibule, similar to the indapamide-bound structure (Response Fig. 4a). In the asymmetrical chlorthalidone-bound structure, S533 points toward the vestibule and downward in TMDs with close and loose CTD interactions, respectively (Response Fig. 4d). The “neutral” influence of chlorthalidone on S533 and the extracellular end of TM10 might explain why chlorthalidone-bound NCC can adopt either dimer configuration, similar to either polythiazide-bound or indapamide-bound structures.

We have now included these structural analyses and relevant discussion in the main text (page 8, line 237-255 and Fig. 5 and Supplementary Fig. 9a,b).

Response Fig. 4 Differential NCC dimer configurations related to ligand-dependent interactions with S533 in TM10. PTZ: polythiazide.

4. The authors show that the residues N149 and N227 are critical for the chlorthalidone, polythiazide, and indapamide inhibition of NCC since the substitution of these residues with alanine decreased the IC₅₀ for these diuretics. However, in previous work (*Sci. Adv.* 8 (45), eadd7176. 2022) it was shown that these substitutions result in loss of the NCC function. Thus, the authors must show the I125 uptake in wild type and N149A,N227A mutant NCC to demonstrate to what extent the NCC function is preserved.

Response:

In our radioactive iodide uptake assay using HEK293S cells stably expressing different NCC variants, N149A substitution did not impair NCC transport function (Response Fig. 5). For N227A substitution, the transport activity of NCC(N227A) is about 20% of that of NCC(WT) (Response Fig. 5). We have now included transport activity of hNCC(N149A) and hNCC(N227A) in Supplementary Fig. 9c.

There are several differences between the functional assays used by Nan et al. (PMID: 36351028) and ours. Nan et al. employed HEK293T cells transiently transfected with NCC constructs N-terminally fused with a halide-sensitive eYFP, with transport activity being measured by iodide-induced quenching of eYFP fluorescence. In contrast, we used HEK293S cells stably expressing unmodified hNCC constructs and measured radioactive iodide uptake, which offer more uniform expression, higher expression levels, and wider linear detection limits. In addition, other assay conditions (such as uptake time and substrate concentrations) might also contribute to differences in functional results.

Response Fig. 5 Functional effects of N149A and N227A substitutions. The values are normalized to that of NCC(WT). Data are shown as mean \pm s.d. ($n = 6$ independent experiments). The P -value was derived from one-way ANOVA with multiple comparisons test corrected by the Tukey method.

5. In this regard, Moreno et al. (*JBC*, vol.291, No.43. 2016 and *AJP Cell Physiol* 323: C385-C399, 2022) showed that βNCC from eel is a NaCl cotransporter that is not sensitive to thiazides, but most of the residues suggested in this work are present in this orthologue. Could the authors revise the eel βNCC sequence and make comments about the potential explanations for not being sensitive to thiazides, although the residues are conserved?

Response:

Thank you for the suggestions. Sequence analysis revealed four thiazide-interacting hNCC (αNCC) residues that differ in eel βNCC ($\text{eNCC}\beta$): G230^{hNCC} (A248 in eel βNCC), H234^{hNCC} (N252), T352^{hNCC} (I370), and C472^{hNCC} (F491). Functional studies show that hNCC with substitution G230A, H234N, or T352I can still be fully inhibited by 100 μM polythiazide (Response Fig. 6), indicating they are not the major contributors to $\text{eNCC}\beta$'s thiazide insensitivity. In contrast, the C472F mutant can only be partially inhibited by 100 μM polythiazide (~50%) (Response Fig. 6), which suggests that C472F substitution decreases NCC sensitivity to polythiazide by

~200,000-fold (IC_{50} increases from ~0.5 nM to ~100 μ M). This substantial effect suggests that F491 in eNCC β (corresponding to C472 in hNCC) is probably the major cause of thiazide insensitivity.

Structural analysis revealed that the bulky phenylalanine side chain in C472F substitution would cause substantial steric clashes with polythiazide, chlorthalidone, and indapamide (Response Fig. 7). For polythiazide, the modeled C472F would clash with the benzothiadiazine moiety. Since benzothiadiazine is a shared structure of thiazide-type diuretics, C472F is expected to generate substantial steric clashes with other thiazide-type diuretics. We have included potential explanations for eel β NCC's thiazide insensitivity in the manuscript (page 10-11, line 336-351 and Supplementary Fig. 8g-j).

Response Fig. 6 Functional effects of hNCC thiazide-interacting residues that are not conserved in eNCC β . The values are normalized to that of hNCC(WT) in the absence of polythiazide. Data are shown as mean \pm s.d. (For control and hNCC(WT), $n = 8$ and 11 independent experiments, respectively. For N234N, T352I, and C472F, $n = 4$ independent experiments. For G230A, $n = 3$ independent experiments). The P -value was calculated via two-sided unpaired t test (Holm-Šídák method for multiple testing correction).

Response Fig. 7 The model of C472F substitution-induced steric clashes with different thiazide diuretics. The models of substituted phenylalanine residues are

shown as dashed outlines. The distances between substituted phenylalanine residues and individual compounds are shown as cyan dashed lines.

6. Also, residues like H234, T352, and C472 that are suggested to be critical for the Na-Cl transport are not present in eel β NCC sequence which however perform as Na-Cl cotransporter.

Response:

Our conclusion that H234, T352, and C472 are critical for hNCC transport function is based on alanine substitution studies. In the eel β NCC (eNCC β), the corresponding residues are N252, I370, and F491, respectively. Notably, hNCC variants with these eNCC β -specific substitutions (H234N, T352I, or C472F) remain functional (Response Fig. 6). Therefore, although these residues are not conserved in eNCC β , the corresponding residues in eNCC β can still support eNCC β to function as a Na-Cl cotransporter. Notably, although T352A, and C472A substitutions severely reduced hNCC transport function to background levels (Response Fig. 8a), hNCC(H234A) was still functional in our radioactive iodide uptake assay (Response Fig. 8b). Therefore, H234 is important but not essential for hNCC transport function.

Response Fig. 8 Functional effects of alanine substitutions of hNCC H234, T352, and C472. The values are normalized to that of hNCC(WT) in the absence of corresponding thiazide diuretics. Data are shown as mean \pm s.d. ($n = 4$ independent experiments). The P -value was calculated via two-sided unpaired t test (Holm-Šidák method for multiple testing correction).

7. Regarding the single nucleotide polymorphisms in NCC, in addition to V229M and V231M, could the authors provide a table showing SNPs located in the critical thiazide and thiazide-like binding sites and comment on potential effects?

Response:

Thank you for the suggestion. We have now prepared a table to include SNPs in the thiazide binding site and commented on their potential impact (Supplementary Table 2). This provides additional insights into SNP of hNCC.

Reviewer #2 (Remarks to the Author):

The manuscript investigates how thiazide-like diuretics, chlorthalidone and indapamide, inhibit the human Na-Cl cotransporter (hNCC). By combining structural and functional studies, the research identifies both commonalities and differences in inhibition mechanisms between these and thiazide-type diuretics. The structural analysis uncovers genetic variations in hNCC that influence diuretic potency, offering insights into NCC's molecular pharmacology and suggesting avenues for precision medicine in hypertension management. Overall, this manuscript is well prepared. However, some revisions are needed:

We appreciate the reviewer's positive comments. We have followed the reviewer's advice to address specific comments. We believe that these revisions have significantly strengthened the manuscript.

1. *Comparing Supplementary Fig. 3f and Fig. 4g, it appears that Fig. 4g lacks some angular distributions, which might affect the density quality in the chlorthalidone-bound hNCC structure. Also, the TMD is missing in the final map, given the small data set, could additional data collection and 3D variability analysis address this issue?*

Response:

We appreciate the reviewer's suggestion. In the revision, we have focused on a functional hNCC construct with E240A mutation, replacing the nonfunctional double-mutant construct (S345E/L525C) to better address the functionality of the construct for structural studies (Response Fig. 1a). In our functional assay, NCC_{chimera}(E240A) shows comparable sensitivity to chlorthalidone and indapamide as the wild-type hNCC (Response Fig. 1b). To improve the map quality and address the issue of limited angular distributions, we have followed the advice and collected larger datasets and conducted thorough data processing, which indeed significantly improved the structures of chlorthalidone-bound hNCC (Response Fig. 2). As shown in the updated Supplementary Figures 4g/4j, the angular distribution has significantly improved.

Response Fig. 1 Functional effects of E240A and S345E/L525C. **(a)** The values are normalized to that of hNCC(WT) in the absence of metolazone (MTZ). Data are shown as mean \pm s.d. ($n = 4$ independent experiments). **(b)** Data are shown as mean \pm SEM ($n = 3$ independent experiments).

Response Fig. 2 Structures of NCC_{chimera}(E240A) in complex with chlorthalidone.

Furthermore, we were able to resolve two different conformations of chlorthalidone-

bound NCC (Response Fig. 2). One structure (3.01 Å resolution) adopts a symmetrical dimer configuration with widely separated TMD subunits, similar to the indapamide-bound structure (Response Fig 2a). Another structure (3.24 Å resolution) adopts an asymmetrical dimer with contacting TMD subunits and a tilted CTD, similar to the polythiazide-bound structure (Response Fig. 2b), though the TMD subunit forming loose CTD contacts is less well resolved, indicating greater intradimer dynamics of this TMD relative to the rest part of protein. Indeed, 3D variability analysis significantly improved the density of the less CTD-interacting TMD subunit, albeit slightly lower overall resolution (Response Fig. 3). We have now included the results of 3D variability analysis in Supplementary Fig. 5d. The chlorthalidone binding pose remains identical to what we observed in the double mutant. Together, these new data provided significantly improved structural insights.

Response Fig. 3 3D variability analysis of NCC_{chimera}(E240A) in complex with chlorthalidone.

2. *In analyzing the indapamide-binding site, water-mediated interactions are crucial. Binding assays should include T352A to assess how substituting alanine affects hNCC sensitivity to indapamide inhibition.*

Response:

Thank you for pointing out the role of T352 in water-mediated interactions between hNCC and indapamide. In our radioactive iodide uptake assay, T352A reduced hNCC transport activity to background levels (Response Fig. 4), with no detectable thiazide-sensitive iodide uptake activity. As a result, we were not able to assess the functional effects of T352A substitution on hNCC sensitivity to indapamide inhibition. We have now included the functional data (Supplementary Fig. 9d) and added a

sentence in the text (page 8, line 258-262) to clarify this.

Response Fig. 4 Functional effects of T352A substitution. The values are normalized to that of hNCC(WT) in the absence of polythiazide. Data are shown as mean \pm s.d. (n = 4 independent experiments).

3. *The Single Nucleotide Polymorphism Database identifies G230D as a disease-causing mutation. Since the article only covers G230A, including data on G230D would provide a more comprehensive analysis.*

Response:

We agree on the importance of including data on functional effects of G230D mutation. This mutation severely impaired hNCC transport activity to background levels, with no detectable thiazide-sensitive iodide uptake activity (Response Fig. 5). Its negative functional effect is consistent with its association with Gitelman syndrome (PMID: 17654016). However, due to the severe functional impairment, we were not able to assess its impact on hNCC sensitivity to different thiazide diuretics. We have now included the functional data on G230D in the manuscript (page 9, line 275-278; Supplementary Fig. 9d).

Response Fig. 5 Functional effects of G230D mutation. The values are normalized to that of hNCC(WT) in the absence of polythiazide. Data are shown as mean \pm s.d.

(n = 4 independent experiments).

4. *Although hNCC with indapamide and chlorthalidone shows a similar overall structure to the polythiazide-bound form, differences in the distance between TMD subunits are evident. It is essential to label the distance between the TMDs in Fig1b and measure the RMSD to understand the underlying causes of the variations between the two drugs.*

Response:

We appreciate the reviewer's comments on the ligand-dependent dimer configurations of hNCC. We have now labeled the distances between the TMD subunits in Fig. 1 and included RMSD between different structure in the text (page 4, line 99-102; page 5, line 116-119 and 122-124). From our larger cryo-EM data set on E240A mutant, we were able to resolve two conformations of chlorthalidone-bound hNCC. One structure adopts a symmetrical dimer configuration with the TMD subunits being widely separated by 22.4 Å (the distance between the C β of W586 pair) and both forming extensive interactions with the CTD (Response Fig. 2a). This symmetrical chlorthalidone-bound structure is very similar to the indapamide-bound structure (RMSD = 0.474 Å) (Response Fig. 6a). The other chlorthalidone-bound structure adopts an asymmetrical dimer configuration with the CTD tilted and forming extensive interactions with one TMD subunit (Response Fig. 2b). In this asymmetric structure, the TMD subunits are in contact with each other with the distance between the C β of W586 pair being 4.4 Å. This asymmetrical chlorthalidone-bound structure is similar to the polythiazide-bound structure (PDB: 8FHN) (RMSD = 0.978 Å) (Response Fig. 6b).

Response Fig. 6 Comparisons between overall structures of hNCC in complex with

different thiazide diuretics. CTN: chlorthalidone; IDP: indapamide; PTZ: polythiazide.

Despite different dimer configurations, the structure of the more CTD-interacting TMD subunit of asymmetrical chlorthalidone-bound hNCC is nearly identical to that of indapamide-bound hNCC (RMSD = 0.382 Å) (Response Fig. 7a). In contrast, the less CTD-interacting TMD subunit of chlorthalidone-bound hNCC shows some noticeable differences to that of indapamide-bound hNCC (RMSD = 0.832 Å), with conformational changes in TM10, intracellular loop 5 (IL5), TM11, and TM12 (Response Fig. 7b).

Response Fig. 7 Structural comparisons between the TMD subunits of indapamide-bound and asymmetrical chlorthalidone-bound hNCC.

The conformational differences in TM10~TM12 between symmetrical (indapamide- or chlorthalidone-bound) and asymmetrical (polythiazide- or chlorthalidone-bound) hNCC may potentially arise from differential protein-ligand interactions, particularly with S533 and the extracellular end of TM10 (Response Fig. 8a). When bound to polythiazide, S533's side chain points away from the central vestibule to accommodate the 3-position bulky substituent; while with indapamide, S533 points toward the vestibule, stabilized by a water-mediated interaction (Response Fig. 8a). These interactions presumably could induce conformational differences in TM10, affecting the subsequent IL5 and TM11/12, which are involved in TMD-CTD interactions and the dimer interface, respectively (Response Fig. 8b and c). As a result, conformational differences in TM10 may contribute to the different NCC dimer configurations. For chlorthalidone, its bulky substituent is farther from TM10 compared with polythiazide, and it does not form stabilizing polar interactions with

S533 as indapamide (Response Fig. 8a). In addition, its shorter structure does not extend beyond S533 (Response Fig. 8a). As a result, chlorthalidone potentially has a “neutral” effect on S533 and the extracellular end of TM10, and the side chain of S533 adopts varied orientations in chlorthalidone-bound structures. In the symmetrical chlorthalidone-bound structure, S533 points toward the vestibule, which is similar to the indapamide-bound structure (Response Fig. 8a). In the asymmetrical chlorthalidone-bound structure, S533 adopts distinct orientations, pointing toward the vestibule in the subunit with greater interaction with CTD and downward in the subunit with lesser CTD interaction (Response Fig. 8d). The “neutral” effect of chlorthalidone on S533 and the extracellular end of TM10 might explain why chlorthalidone-bound NCC can adopt either dimer configuration similar to polythiazide-bound or indapamide-bound structures.

We have now included these structural analyses and the discussion of the proposed potential mechanism in the manuscript (page 8, line 237-255 and Fig. 5 and Supplementary Fig. 5f,g and 9a,b).

Response Fig. 8 Differential NCC dimer configurations related to ligand-dependent interactions with S533 in TM10.

Reviewer #3 (Remarks to the Author):

The manuscript by Lee and colleagues presents two cryo-EM structures of an engineered human NCC transporter in complex with thiazide-like diuretics, specifically chlorthalidone and indapamide. This work is a “sequel” to the authors' previous work on the hNCC transporter¹, offering valuable new insights into the mechanisms of NCC inhibition by thiazide-like and thiazide-type diuretics. The structural and functional data suggest that the shared benzenesulfonamide moieties of these diuretics interact with NCC in similar ways, while the distinct functional groups of each diuretic form unique interactions with the transporter. A notable aspect of this study is the demonstration that polymorphisms could influence NCC's response to different diuretics. The mechanisms uncovered in this work are significant and could contribute to the refinement of existing diuretics for hypertension treatment. While the manuscript is undoubtedly of high importance to the field, I have several concerns and suggestions that should be addressed.

We appreciate the reviewer's positive comments. We have followed the reviewer's advice to address specific comments. We believe these revisions have significantly improved the manuscript.

- 1. It is surprising to see that only one TMD is resolved in chlorthalidone-bound hNCC structure. Given similar binding affinity between indapamide (IC₅₀=9.3 nM) and chlorthalidone (IC₅₀=24 nM) to the hNCC, what could account for the markedly different intradimer dynamics caused by their binding to the transporter?*

Response:

We appreciate the reviewer's suggestion and comment on the potential underlying mechanisms of NCC ligand-specific intradimer dynamics. In the revision, we have focused on a functional hNCC construct with E240A mutation, replacing the nonfunctional double-mutant construct (S345E/L525C) to better address the functionality of the construct for structural studies (Response Fig. 1a). In our functional assay, NCC_{chimera}(E240A) shows comparable sensitivity to chlorthalidone and indapamide as the wild-type hNCC (Response Fig. 1b). To improve the map quality, we have collected larger datasets and conducted thorough data processing (including 3D variability analysis), which significantly improved the structures of chlorthalidone-bound hNCC (Response Fig. 2).

Response Fig. 1 Functional effects of E240A and S345E/L525C. **(a)** The values are normalized to that of hNCC(WT) in the absence of metolazone (MTZ). Data are shown as mean \pm s.d. (n = 4 independent experiments). **(b)** Data are shown as mean \pm SEM (n = 3 independent experiments).

Response Fig. 2 Structures of NCC_{chimera}(E240A) in complex with chlorthalidone.

Interestingly, we were able to resolve two different conformations of chlorthalidone-bound NCC (Response Fig. 2). One structure (3.01 Å resolution) adopts a symmetrical dimer configuration with well-separated TMD subunits, similar to the indapamide-bound structure (Response Fig 2a and 3a). Another structure (3.24 Å resolution) adopts an asymmetrical dimer with contacting TMD subunits and a tilted CTD, similar to the polythiazide-bound structure (Response Fig. 2b and 3b), though the TMD subunit forming loose CTD contacts is less well resolved, indicating greater intradimer dynamics. The chlorthalidone binding pose remains identical to what we observed in the double mutant. The two conformations and intradimer dynamics of chlorthalidone-bound NCC provide a possible explanation to a very weak TMD in the previous structure from smaller datasets, in which these conformational heterogeneities were not resolved.

Response Fig. 3 Comparisons between overall structures of hNCC in complex with different thiazide diuretics.

Structural comparisons between symmetrical (indapamide- or chlorthalidone-bound) and asymmetrical (polythiazide- or chlorthalidone-bound) hNCC reveal conformational differences in TM10~TM12 (Response Fig. 4c), which may potentially arise from differential protein-ligand interactions, particularly with S533 and the extracellular end of TM10 (Response Fig. 4a). When bound to polythiazide, S533's side chain points away from the central vestibule to accommodate the 3-position bulky substituent; while with indapamide, S533 points toward the vestibule, stabilized by a water-mediated interaction (Response Fig. 4a). These interactions presumably could induce conformational differences in TM10, affecting the subsequent intracellular loop 5 (IL5) and TM11/12, which are involved in TMD-CTD interactions

and the dimer interface, respectively (Response Fig. 4b and c). As a result, conformational differences in TM10 may contribute to the different NCC dimer configurations. For chlorthalidone, its bulky substituent is farther from TM10 compared with polythiazide, and it does not form stabilizing polar interactions with S533 as indapamide (Response Fig. 4a). In addition, its shorter structure does not extend beyond S533 (Response Fig. 4a). As a result, chlorthalidone potentially has a “neutral” effect on S533 and the extracellular end of TM10, and the side chain of S533 adopts varied orientations in chlorthalidone-bound structures. In the symmetrical chlorthalidone-bound structure, S533 points toward the vestibule, which is similar to the indapamide-bound structure (Response Fig. 4a). In the asymmetrical chlorthalidone-bound structure, S533 adopts distinct orientations, pointing toward the vestibule in the subunit with greater interaction with CTD and downward in the subunit with lesser CTD interaction (Response Fig. 4d). The “neutral” effect of chlorthalidone on S533 and the extracellular end of TM10 might explain why chlorthalidone-bound NCC can adopt either dimer configuration similar to polythiazide-bound or indapamide-bound structures.

We have now included the new structures, the structural analyses, and the discussion of the proposed potential mechanism in the manuscript (page 5, line 115-128; page 8, line 237-255; Fig. 1 and 5 and Supplementary Fig. 5f,g and 9a,b).

Response Fig. 4 Differential NCC dimer configurations related to ligand-dependent

interactions with S533 in TM10.

2. *Based on the 2D classification results shown in Supplementary Figure 4c, it appears that both TMDs exhibit clear features in some classes. Did the author ever isolate any particles that results in a map with an intact TMD dimer? Please show some reference maps used in 3D classification. Also, why were the particles used for the final map reconstruction (Supplementary Figure 4a) significantly fewer than those in the indapamide-bound dataset?*

Response:

During the revision, we have focused on the functional hNCC construct with E240A mutation. With significantly larger datasets and thorough data processing, we were able to clearly resolve both TMDs and obtained structures of hNCC in complex with chlorthalidone in two different dimer configurations. Now, there are large number of particles for final reconstruction. We suspect that previous very weak density in one TMD might be due to inability to separate multiple conformations from the much smaller dataset. We have now included these new structures in two different configurations (Fig. 1) and also included several reference maps used for 3D classification in Supplementary Figure 4a.

3. *The FSC curves in Supplementary Figure 3e do not converge to 0.*

Response:

Thank you very much for bringing this to our attention. This issue has been resolved by fixing an error in the duplicate removal on this particular dataset. As a result, the FSC curves in the updated Supplementary Figure 3e now converge to 0.

4. *For better comparison, please highlight the NTD in both maps and models shown in Figures 1a and 1b.*

Response:

We have followed the advice and highlighted the NTD in both maps and models shown in Figure 1.

5. *There is no mention of ATP/ADP in the entire manuscript, which is somewhat surprising, given that adenine nucleotide binding was reported to be important in the previous study¹. Are there any ATP/ADP densities present in either of the maps?*

Response:

Thank you for pointing this out. We observed ATP/ADP bound at the same location with the same binding pose as in previous polythiazide-bound structures (Response Fig. 5). We have now added a short description in the text (page 6; line 174-177) and Supplementary Fig. 7d-g to clarify this point.

Response Fig. 5 Nucleotide densities and nucleotide-binding sites in structures of NCC_{chimera}(E240A) in complex with different thiazide diuretics.

6. Line 191. “hydrophobic interactions with G230, M233, I532, and S533”. A figure call is suggested here, as G230 is not depicted until Figure 4a.

Response:

Thank you. We have now highlighted G230 residue in Fig. 3d, illustrating interactions between hNCC and chlorthalidone.

7. Line 205. “For C472, although its side chain interacts with all three molecules, it appears to make the strongest interactions with polythiazide.” What is the supporting evidence?

Response:

In polythiazide-bound structures, there are strong densities connecting the side chain of C472 and the 1-position oxygen of polythiazide (Response Fig. 6a). In contrast, no such strong densities connecting C472 and chlorthalidone or indapamide (Response Fig. 6b and c) were observed. For polythiazide, the functional group facing the side chain of C472 is the 1-position SO₂ group. It appears that both oxygen atoms of the 1-position SO₂ group form polar interactions with C472’s side chain (Response Fig. 6d). However, for chlorthalidone, there is only one hydroxyl group to form polar interactions with C472’s side chain (Response Fig. 6e). Because indapamide is a bit

farther away from C472, C472 does not form polar interaction with indapamide (Response Fig. 6f). We have now added a short description in the text (page 8, line 230-237) and Supplementary Fig. 8a-f to clarify this point.

Response Fig. 6 Interactions between C472 and different thiazide diuretics.

8. *What is the transport activity of the G230A mutant? Additionally, I am curious about how the G230D variant affects transport activity; did the authors test this?*

Response:

The transport activity of hNCC(G230A) is about 90% of hNCC(WT) (Response Fig. 7a). However, G230D mutation severely impaired hNCC transport activity to background levels with no detectable thiazide-sensitive iodide uptake activity (Response Fig. 7b). The negative functional effect of G230D mutation is consistent with its association with Gitelman syndrome (PMID: 17654016). Due to the severe functional impairment, we were not able to assess the effect of G230D on hNCC sensitivity to different thiazide diuretics. We have now included the functional results of G230A and G230D substitutions in the text (page 8, line 265-266; page 9, line 275-278) and Supplementary Fig. 9d,e.

Response Fig. 7 Functional effects of G230A and G230D mutations. The values are normalized to that of hNCC(WT) in the absence of polythiazide. Data are shown as mean \pm s.d. (n = 4 independent experiments).

9. *Can the authors provide insight into why the V229M variant exhibits enhanced transport activity from a structural and mechanistic perspective?*

Response:

As V229M shows comparable surface expression (Response Fig. 8c), as assessed by surface staining method we previously developed (PMID: 36792826), its increased activity is indeed due to enhanced transporter activity. Structural comparisons reveal that the local environment surrounding V229 undergoes substantial changes between outward- and inward-facing conformations (Response Fig. 8a and b). Interestingly, in the inward-facing conformation, V229 faces a narrow space, and its side chain, when substituted with methionine, may require some local conformational re-arrangement to avoid steric clashes with I531 (Response Fig. 8a). In contrast, in the outward-facing conformation, V229 points to an open space, which can easily accommodate a V229M substitution (Response Fig. 8b). As a result, V229M mutation might destabilize the inward-facing conformation, relative to the outward-facing conformation, thus affecting the energy landscape of the conformational transition during the transport cycle. We suspect this might thus impact the transport cycle, potentially leading to the enhanced activity. For SLC12 family members, including NCC, the inward-facing conformation seems to be the most energetically favorable state given all the apo structures were determined in this conformation. It is possible that transition from an inward-facing to an outward-facing conformation might be the major energy barrier during the transport cycle. Nonetheless, future studies are needed to understand the molecular determinants that govern the transport rate of NCC. We have edited the text and included our hypothesis about why this mutant might potentially lead to increased activity (page 9,

line 286-291; Supplementary Fig. 9i).

Response Fig. 8 Potential underlying mechanisms of increased hNCC transport activity caused by V229M. (c) The values are normalized to that of hNCC(WT). Data are shown as mean \pm s.d. (n = 3 independent experiments).

10. To better understand the effects of diuretics on different mutants, please provide the IC₅₀ values from the dose-response curves.

Response:

The IC₅₀ values have now been included in the figures to facilitate quantitative comparison.

11. Why were different detergents used during the SEC step? For the chlorthalidone-bound sample, detergent exchange to GDN was performed, whereas LMNG/CHS was used throughout the preparation for the indapamide-bound sample. Please provide a rationale for this choice.

Response:

Different detergents were used during the development of the project. The chlorthalidone-bound NCC_{chimera}(S345E/L525C) was initially prepared for both detergent (GDN) and nanodisc samples, following our experiences with polythiazide-bound hNCC. In contrast, for indapamide-bound NCC_{chimera}(S345E/L525C), as we found nanodiscs worked better, we focused directly on nanodisc preparation, keeping the sample in LMNG/CHS without switching to GDN before reconstitution.

To avoid any potential differences caused by different detergents, the new chlorthalidone- and indapamide-bound NCC_{chimera}(E240A) nanodisc samples were all prepared following the same procedure only using LMNG/CHS throughout the preparation.

12. *Regarding cryo-EM sample preparation, 3 μ l of the protein sample was applied to the grids. Are any surfactants required during this step?*

Response:

No surfactant was applied when preparing cryo-EM grids.

REVIEWERS' COMMENTS

Reviewer #1 (Remarks to the Author)

I agree with all changes made by the authors.

In the original description of the polythiazide binding site the G320 was not included and the A529 was included. But not, G320 was mentioned as part of the site, while A529 is not. Please clarify.

Response:

Thank you very much for the positive comments and for pointing out the ligand-specific binding site differences involving G230 and A529. G230 contributes to interactions with chlorthalidone, but not with polythiazide. Therefore, G230 was not included in the original description of the polythiazide-binding site. In contrast, A529 contributes to interactions with polythiazide, but not with chlorthalidone or indapamide, two commonly used thiazide-like diuretics. Since this manuscript focuses on thiazide-like diuretics, A529 is not described as part of the binding site. We appreciate the reviewer's time and effort during the review process.

Reviewer #2 (Remarks to the Author):

Thank you for the excellent revisions. All my questions have been addressed, and I have no further comments.

Response:

We appreciate the favorable comments and the reviewer's time and effort during the review process.

Reviewer #3 (Remarks to the Author):

TheThe manuscript has undergone significant improvement, and all of my concerns have been addressed. Congratulations to the authors for their exceptional work.

Response:

We appreciate the favorable comments and the reviewer's time and effort during the review process.